# From Cradle to Cane: A Two-Pass Framework for High-Fidelity Lifespan Face Aging

**Tao Liu**[1], **Dafeng Zhang**[2], **Gengchen Li**[3], **Shizhuo Liu**[2], **Yongqi Song**[2]
**Senmao Li**[1], **Shiqi Yang**[1,*] **Boqian Li**[4], **Kai Wang**[5,6,7], **Yaxing Wang**[8†]

[1]VCIP, College of Computer Science, Nankai University
[2]Samsung Research China - Beijing (SRC-B)
[3]School of Electrical and Information Engineering, Zhengzhou University
[4]School of Computer , Zhengzhou University of Aeronautics
[5]Program of Computer Science, City University of Hong Kong (Dongguan)
[6]City University of Hong Kong [7]Computer Vision Center, Barcelona
[8]College of Artificial Intelligence, Jilin University

{lt0lcy0, senmaonk, shiqi.yang147.jp, 1602522393boxili}@gmail.com
{dfeng.zhang, shizhuo.liu, yongqi.song}@samsung.com
{lgc204747899}@gs.zzu.edu.cn, {kai.wang}@cityu-dg.edu.cn,
{yaxing}@nankai.edu.cn

## Abstract

Face aging has become a crucial task in computer vision, with applications ranging from entertainment to healthcare. However, existing methods struggle with achieving a realistic and seamless transformation across the entire lifespan, especially when handling large age gaps or extreme head poses. The core challenge lies in balancing *age accuracy* and *identity preservation*—what we refer to as the *Age-ID trade-off*. Most prior methods either prioritize age transformation at the expense of identity consistency or vice versa. In this work, we address this issue by proposing a *two-pass* face aging framework, named *Cradle2Cane*, based on few-step text-to-image (T2I) diffusion models. The first pass focuses on solving *age accuracy* by introducing an adaptive noise injection (*AdaNI*) mechanism. This mechanism is guided by including prompt descriptions of age and gender for the given person as the textual condition. Also, by adjusting the noise level, we can control the strength of aging while allowing more flexibility in transforming the face. However, identity preservation is weakly ensured here to facilitate stronger age transformations. In the second pass, we enhance *identity preservation* while maintaining age-specific features by conditioning the model on two identity-aware embeddings (*IDEmb*): *SVR-ArcFace* and *Rotate-CLIP*. This pass allows for denoising the transformed image from the first pass, ensuring stronger identity preservation without compromising the aging accuracy. Both passes are *jointly trained in an end-to-end way*. Extensive experiments on the CelebA-HQ test dataset, evaluated through Face++ and Qwen-VL protocols, show that our *Cradle2Cane* outperforms existing face aging methods in age accuracy and identity consistency. Additionally, *Cradle2Cane* demonstrates superior robustness when applied to in-the-wild human face images, where prior methods often fail. This significantly broadens its applicability to more diverse and unconstrained real-world scenarios. Code is available at https://github.com/byliutao/Cradle2Cane.

---

*:visiting researcher in Nankai University

†:Corresponding author

39th Conference on Neural Information Processing Systems (NeurIPS 2025).

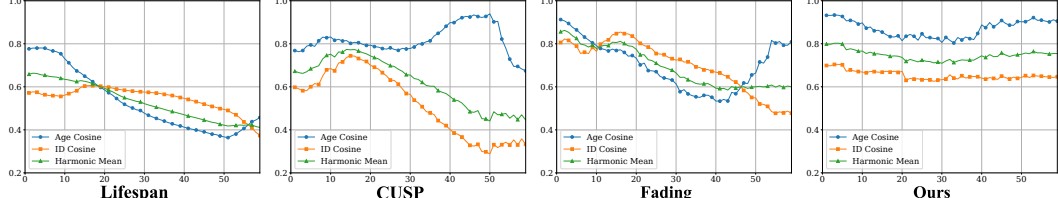

Figure 1: *Age–ID trade-off* curves across sixty age shift values. We compute the Age/ID cosine similarities over 100 human faces across 1-60 age shift values and the corresponding harmonic means. Existing approaches tend to favor either age accuracy or identity consistency, resulting in imbalanced performance across the entire lifespan ages. In contrast, our method *Cradle2Cane* achieves a better balance between the two objectives. More details and results are provided in Appendix A.5.

# 1   Introduction

Deep learning [34] has allowed a realistic alteration of the apparent age of a person [7, 12, 80], opening promising applications in areas such as computer graphics, entertainment, forensics and healthcare. The goal of facial age transformation is to simulate the natural aging or de-aging process in a visually convincing manner. To this end, numerous methods have been developed to achieve high-quality, identity-preserving age progression and regression. Recent approaches are based on deep generative models, such as generative adversarial networks (GANs) [1, 15, 22] and Diffusion Models (DMs) [3, 7, 26, 69, 75], and have shown promising results. However, to the best of our knowledge, most existing methods suffer from a limited transformation range and often struggle to maintain *high-fidelity* results when handling large age gaps, occlusions, or extreme head poses. As a result, they fall short of delivering seamless *cradle-to-cane* face aging performance across the entire lifespan.

In this study, we attribute the limitations of existing face aging methods to an imbalanced trade-off between *age accuracy* and *identity consistency*—a challenge we term the *Age-ID trade-off*. Most prior approaches [1, 7, 26] tend to emphasize one aspect while neglecting the other due to their unified framework to deal with entire lifespan ages, resulting in either visually convincing age transformations that compromise identity, or identity-preserving outputs with inaccurate aging effects. This imbalance is evident in the trade-off curves shown in Fig. 1, where, for example, as the age difference increases, methods such as FADING [7], CUSP [14], and Lifespan [48] tend to show improved aging realism at the cost of reduced identity preservation, or vice versa. The fluctuating curves of harmonic means further demonstrate this phenomenon. To address the *Age-ID trade-off* problem, we propose our face aging framework built upon few-step T2I diffusion models *SDXL-Turbo* [59], which offer two key advantages: 1) the few-step nature of these models [11, 35, 40, 59] enables fast inference while maintaining high image *fidelity*, and 2) the flexible noise control in the forward diffusion process allows fine-grained modulation of aging strength by adjusting the injected noise scale. As illustrated in Fig. 2, injecting higher noise levels during the forward diffusion process increases editability, enabling more pronounced aging transformations while downgrading the identity consistency. In contrast, lower noise levels better preserve identity information with less aging accuracy, highlighting the trade-off between visual age change and identity consistency. Similar identity-editing trade-offs are also observed in image editing and generation [19, 36, 43, 70, 67, 24]. However, directly applying the few-step diffusion models cannot achieve fine-grained face aging with identity consistency and age accuracy, resulting from that the few-step T2I diffusion models [20, 53, 63, 38] do not inherently support age or identity conditions.

In this paper, we propose to address the *Age-ID trade-off* by decoupling age accuracy and identity preservation into a *two-pass*[‡] diffusion framework *Cradle2Cane*, with *SDXL-Turbo* as the backbone and each stage is tailored to optimize a specific objective. During the *first* pass, which focuses on precise age control, we introduce an adaptive noise injection (*AdaNI*) mechanism guided by textual descriptions which containing age and gender attributes. The level of noise injected is dynamically adjusted based on the magnitude of the desired age transformation. Naturally, human identity is better preserved with smaller age variations and tends to degrade with larger age gaps. This strategy aims to overcome the limitations of existing methods that rely on uniform solutions for modeling aging across the entire lifespan. In this stage, identity is only weakly preserved to allow greater flexibility

---

[‡]We define a *pass* as the T2I inference process that generates a real image through the diffusion model.

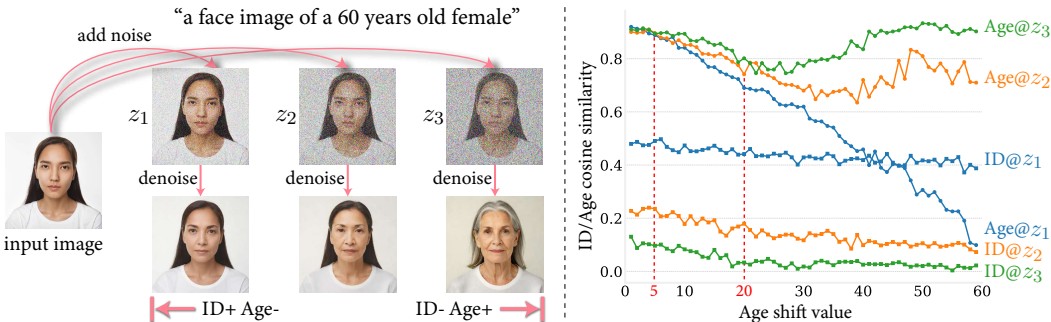

Figure 2: (Left) We illustrate the effects of injecting three different levels of noise into the input image, as used in the 4-step SDXL-Turbo image-to-image pipeline. As visually evident, higher noise levels lead to more pronounced age transformations at the cost of reduced identity preservation. (Right) We present a statistical analysis on 100 human faces, that quantitatively demonstrates the Age-ID trade-off inherent in face aging tasks. Specifically, we evaluate three representative noise injection levels and measure their corresponding impacts on age accuracy and identity consistency.

in age manipulation. In the *second* pass, we reinforce the identity consistency of the generated face while preserving the age-specific characteristics from the first pass. We propose to achieve this by conditioning the few-step diffusion model with a concatenation of two identity-aware embeddings (*IDEmb*): an *SVR-ArcFace* embedding and a *Rotate-CLIP* embedding. These embeddings guide the model to denoise a minimally perturbed input, ensuring stronger identity preservation without compromising the age transformation. It is worth noting that both stages are *jointly trained* in an *end-to-end* manner, where the output image from the first stage is further diffused and used as the noisy latent input for the second stage. After training, our proposed method, *Cradle2Cane*, is capable of achieving *high-fidelity* and adaptive face aging while maintaining a superior balance between age accuracy and identity consistency compared to existing approaches.

In our experiments, we conduct comprehensive comparisons against a diverse set of GAN-based and diffusion-based face aging methods on the CelebA-HQ [27, 41] test dataset. We adopt both the Face++ [14] and Qwen-VL [68] evaluation protocols to assess performance in terms of age accuracy and identity consistency. Both evaluation pipelines consistently validate the effectiveness of our method, *Cradle2Cane*, which achieves a superior balance in the *age-ID trade-off* with inference speeds comparable to GAN-based methods. Furthermore, benefiting from the strong generative capacity of text-to-image diffusion models, *Cradle2Cane* exhibits enhanced robustness on in-the-wild human face images—where previous approaches often struggle—thereby significantly broadening its range of practical application scenarios. To summarize, this paper makes the following main contributions:

- We propose a novel two-pass approach *Cradle2Cane* that decouples *age accuracy* and *identity preservation* in face aging, where the first pass applies adaptive noise injection (*AdaNI*) for precise age manipulation, and the second pass reinforces identity consistency through identity-aware embedding (*IDEmb*).

- For the first pass, we introduce a text-guided adaptive noise injection (*AdaNI*) strategy that dynamically adjusts the injected noise level based on the desired age transformation strength, enabling fine-grained control over the *age-ID trade-off*.

- To enhance identity preservation, we design a conditioning mechanism that leverages a combination of *SVR-ArcFace* and *Rotate-CLIP* as identity-aware embeddings (*IDEmb*), guiding the *second-pass* denoising process for high-fidelity and identity-consistent outputs.

- Extensive experiments on the CelebA-HQ test dataset demonstrate that *Cradle2Cane* consistently outperforms existing baselines across age accuracy, identity consistency and image quality, while maintaining fast inference speed. Moreover, *Cradle2Cane* exhibits strong generalization to in-the-wild human face images—a challenging scenario where current methods often fail.

## 2 Related Work

**Face Aging.** Facial age editing aims to simulate the natural process of fine-grained aging in facial images while faithfully preserving the subject's identity. Traditional approaches relied on physical modeling [52, 66] or attribute manipulation [30, 64], but often struggled with generalization and photorealism. The emergence of GAN-based methods such as Lifespan [48], IPCGAN [72], and CAAE [80] significantly improved aging realism by learning conditional generative mappings from large-scale datasets. For instance, SAM [1] combines an aging encoder with an inversion encoder to perform age transformations in the latent space of StyleGAN2. CUSP [14] disentangles style and content using dual encoders and manipulates them for personalized age transformations. HRFAE [76] introduces an age modulation network that fuses age labels into latent representations to guide high-resolution age progression. With the recent advancements in diffusion models [21, 65], they have emerged as powerful alternatives for high-fidelity face aging [7, 26, 31]. For example, FADING [7] fine-tunes a pretrained LDM [53] on age-labeled datasets. During inference, it uses NTI [19, 44] to embed input images into latent space, allowing for localized age edits. Similarly, IPFE [3] combines latent diffusion with biometric and contrastive losses to enforce identity preservation during facial aging and de-aging. However, we posit that face aging should adhere to a *natural principle*: subtle age variations preserve facial identity more effectively, whereas significant age gaps introduce greater identity distortion, while the current approaches generally overlook this consideration.

**Semantic Latent Spaces in Generative Models.** Linear latent space models of facial shape and appearance were extensively studied in the 1990s, primarily through PCA-based representations [5, 8, 56]. However, these early approaches were limited to aligned and cropped frontal facial images. Afterwards, the StyleGAN-family generative models [28, 29], have demonstrated powerful editing capabilities, largely attributed to the structured and interpretable nature of their latent spaces. In contrast, diffusion models lack an explicit latent space by design. Nevertheless, recent studies have attempted to uncover GAN-like latent structures within them, targeting various representations such as the UNet bottleneck [18, 33, 49, 74], the noise input space [10], and the text embedding space [4]. For example, Concept Sliders [13] propose semantic image editing through low-rank adaptation in weight space, guided by contrastive image or text pairs. Despite these advances, existing disentanglement-based methods are typically limited to coarse-grained attribute control—such as adjusting age, hair, or expression via semantical direction controls—and often struggle to achieve precise, fine-grained manipulation of facial aging features.

**Text-to-Image Models Distillation.** Text-to-image (T2I) models based on diffusion [2, 9, 54, 58] have achieved impressive progress in generating high-quality images from text prompts. Despite their success, the inference phase remains a bottleneck—diffusion models require iterative denoising. To mitigate this, a variety of acceleration methods have been proposed. While training-free approaches have shown promise for both diffusion [25, 35, 42, 81], the most effective strategies often involve additional distillation process to accelerate the sampling process beyond the capabilities of the original base models. SD-Turbo [59] introduces a discriminator combined with a score distillation loss to improve performance. Most of these methods depend on image-text pair datasets for training, requiring substantial data alignment between visual and textual features. In contrast, SwiftBrush [47] adapts variational score distillation. SwiftBrush [47] achieves the first *image-free* training by using generated images as the training set, avoiding the need for paired datasets. In this paper, we build our method, *Cradle2Cane*, upon SDXL-Turbo [59], which is widely adopted and demonstrate strong performance in few-step high-quality image generation, to introduce the two-pass architecture *Cradle2Cane* tailored for controllable facial age transformation.

## 3 Method

In this section, we present our framework *Cradle2Cane* for face aging. Given a source face image $\mathbf{x}_a$ of a person at source age $a$, and a target age $b$, our goal is to generate a realistic target face image $\mathbf{x}_b$, depicting the same identity at age $b$. The main challenge lies in achieving realistic aging effects while preserving the identity (ID) of the subject. Due to the scarcity of datasets containing the same identity across a wide age range, directly transforming $\mathbf{x}_a$ to $\mathbf{x}_b$ remains a difficult task. The full pipeline of our method *Cradle2Cane* is visualized in Fig. 3. We first introduce the preliminaries in Section 3.1. Then Section 3.2 presents the adaptive noise injection (*AdaNI*) during the first pass.

Section 3.3 details the second pass with identity-aware embedding (*IDEmb*) for identity preservation. Section 3.4 defines the training objectives and loss functions.

## 3.1 Preliminary

**Fast Sampling of T2I diffusion models.** SDXL-Turbo [59] accelerates standard diffusion models [51, 55] via *Adversarial Diffusion Distillation*, enabling high-quality image generation in only a few steps. Unlike DDPM [21] or DDIM [65], which typically require 50 to 1000 inference steps, SDXL-Turbo achieves *1-4* steps generation by training a compact denoiser to imitate a large teacher model, supervised jointly by distillation and adversarial losses. The forward process perturbs an initial latent variable $\mathbf{z}_0 \in \mathbb{R}^d$ into increasingly noisy states $\mathbf{z}_1, \ldots, \mathbf{z}_T$ using a Markov chain:

$$q(\mathbf{z}_{1:T} \mid \mathbf{z}_0) = \prod_{t=1}^{T} q(\mathbf{z}_t \mid \mathbf{z}_{t-1}), \tag{1}$$

where $T$ is the denoise steps. The reverse process then reconstructs $\mathbf{z}_0$ from $\mathbf{z}_T$ in $T$ learned steps:

$$p_\theta(\mathbf{z}_{0:T}) = p(\mathbf{z}_T) \prod_{t=1}^{T} p_\theta(\mathbf{z}_{t-1} \mid \mathbf{z}_t), \tag{2}$$

where each $p_\theta(\mathbf{z}_{t-1} \mid \mathbf{z}_t)$ is a Gaussian parameterized by a neural network trained to approximate the inverse of the forward noising process.

**Overall pipeline of our method *Cradle2Cane*.** To address the challenge of controllable and identity-preserving face aging, we propose a two-pass framework, *Cradle2Cane*, built upon the efficient SDXL-Turbo model. In the first stage, we perform adaptive noise injection (*AdaNI*) on the input face image $\mathbf{x}_a$, guided by age-specific embedding, to generate an intermediate image $\hat{\mathbf{x}}_b$ that reflects the target age $b$. This step aims to synthesize realistic aging effects while maintaining essential identity traits. However, for large age gaps, $\hat{\mathbf{x}}_b$ may exhibit partial identity drift due to the strong age transformation. To compensate for this, the second stage focuses on enhancing identity consistency. A lower magnitude of noise is injected into $\hat{\mathbf{x}}_b$, and identity-aware embeddings (*IDEmb*) conditioning is applied using features extracted from the original input image. This results in the final output face image $\mathbf{x}_b$, which exhibits both faithful aging effects and high identity preservation.

## 3.2 *1st Pass:* Adaptive Noise Injection (*AdaNI*) for Age Accuracy

We address the *Age-ID trade-off* by focusing on two critical aspects: *age accuracy* and *identity preservation*. Empirically, we observe that the extent of facial modifications required during age progression is closely correlated with the magnitude of the age gap. Specifically, larger age transitions typically demand more pronounced structural and textural changes, while smaller transitions involve only minor appearance adjustments. Prior works [43], as well as the left portion of Fig. 2, suggest that the level of noise injected into the input image controls the flexibility and intensity of editing.

Building on this insight, we conduct a systematic study to examine how varying noise levels influence the balance between identity fidelity and age realism. In particular, we apply three levels of noise injection, denoted as $z_1$, $z_2$, and $z_3$, to a set of 100 face images under age transformation tasks spanning age shifts from 1 to 60 years. As illustrated in Fig. 2 (right), lower noise injection intensity ($z_1$) consistently leads to superior identity preservation across all age shifts. However, it fails to deliver accurate age progression, particularly for larger age gaps. Conversely, higher noise levels ($z_3$) produce more realistic age transformations but significantly compromise identity consistency. These results highlight a clear trade-off between identity preservation and age accuracy, governed by the noise injection intensity. Motivated by these findings, we propose an adaptive noise injection (*AdaNI*) strategy that dynamically modulates the noise level based on the target age shift.

More specifically, we encode a predefined *age prompt* using the CLIP text encoder to obtain the text embedding, which conditions the generation process via cross-attention. For *AdaNI* injection, we divide the age transformation magnitude into three categories, using ages 5 and 20 as boundaries based on our quantitative analysis in Fig. 2, where age accuracy drops significantly beyond these thresholds.

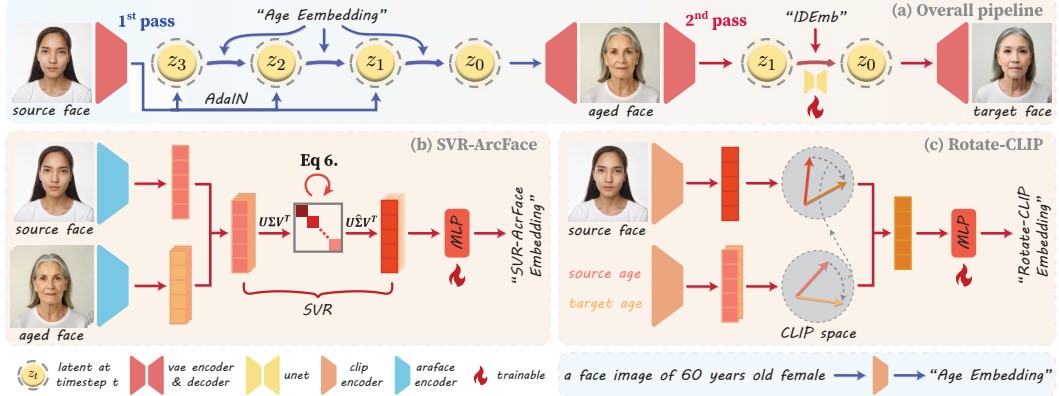

Figure 3: Our method *Cradle2Cane* consists of two passes: the first pass employs adaptive noise injection (*AdaNI*) to enhance age accuracy, while the second pass incorporates identity-aware embeddings (*IDEmb*), including SVR-ArcFace and Rotate-CLIP embeddings, to improve identity consistency. During training the MLPs and UNet-LoRA modules, we jointly optimize identity loss between source and target face images, as well as age and quality losses over the target images.

Each category corresponds to a specific noise level applied during first-pass noise injection:

$$\hat{\mathbf{z}}_0 = \begin{cases} p_\theta(\mathbf{z}_0 \mid \mathbf{z}_1), & |\Delta\text{age}| \leq 5, \\ p_\theta(\mathbf{z}_0 \mid \mathbf{z}_2), & 5 < |\Delta\text{age}| \leq 20, \\ p_\theta(\mathbf{z}_0 \mid \mathbf{z}_3), & |\Delta\text{age}| > 20, \end{cases} \tag{3}$$

where $\mathbf{z}_1, \mathbf{z}_2, \mathbf{z}_3$ represent different noise levels injected into the latent space. After completing the diffusion process to obtain the final latent code $\hat{\mathbf{z}}_0$, the intermediate aged face is reconstructed via the VAE decoder $D$: $\hat{\mathbf{x}}_b = D(\hat{\mathbf{z}}_0)$, which exhibits high age accuracy but relatively weak identity preservation. This enables our model to better balance the competing objectives of age accuracy and faithful identity preservation. Nonetheless, even with adaptive injection, identity degradation becomes increasingly prominent with larger age transitions. To mitigate this effect, the second pass of *Cradle2Cane* explicitly enhances identity consistency by refining identity-specific embeddings.

### 3.3 *2nd Pass:* **Identity-Aware Embedding (*IDEmb*) for Identity Preservation**

To further improve identity preservation, we extract identity-aware embeddings (*IDEmb*) from the source face $\mathbf{x}_a$ using both ArcFace and CLIP encoders, which are standard features [1, 3, 61] for measuring and guiding identity information. A central challenge in this process is the inherent entanglement between age and identity within these embeddings—both ArcFace and CLIP features tend to encode age-related cues alongside identity information [17, 62]. To overcome this limitation, we propose two novel embedding modules: **SVR-ArcFace** and **Rotate-CLIP**. These modules are designed to explicitly suppress age-related components within their respective embedding spaces, thereby disentangling identity from age.

#### 3.3.1 SVR-ArcFace

Given a source face image $\mathbf{x}_a$, we generate a set of $n$ aged face images $\{\mathbf{x}_b^{(i)}\}_{i=1}^n$ by injecting different noise levels in the first stage. These images share the same identity as $\mathbf{x}_a$ but exhibit different age characteristics. Inspired by prior works [16, 37, 39], which suggest that applying Singular Value Decomposition (SVD) followed by singular value reweighting (SVR) can enhance shared features while suppressing divergent ones such as age, we propose a singular value reweighting technique to refine identity features from the ArcFace embeddings. We refer to this method as **SVR-ArcFace**.

First, we extract ArcFace embeddings $u_a$ and $\{u_b^{(i)}\}_{i=1}^n$ from the source and aged face images, and concatenate them into a matrix:

$$U = [u_a, u_b^{(1)}, u_b^{(2)}, \ldots, u_b^{(n)}] \in \mathbb{R}^{D \times (n+1)}, \tag{4}$$

where $D$ is the embedding dimension. We then perform Singular Value Decomposition (SVD) on $U$:

$$U = \mathbf{U}\mathbf{\Sigma}\mathbf{V}^T, \quad \mathbf{\Sigma} = \mathrm{diag}(\sigma_0, \sigma_1, \ldots, \sigma_r), \tag{5}$$

where $r = \min(D, n+1)$. Following the assumption in previous works, we treat the dominant singular values of $U$ as encoding the shared identity, since all embeddings in $U$ correspond to the same person. To suppress age-related variations and emphasize identity features, we apply a nonlinear function for the singular value reweighting (SVR):

$$\hat{\sigma}_i = \beta e^{\alpha \sigma_i} \cdot \sigma_i, \tag{6}$$

where $\alpha, \beta > 0$ are hyperparameters that control the enhancement strength. The reweighted singular value matrix is defined as $\hat{\mathbf{\Sigma}} = \mathrm{diag}(\hat{\sigma}_0, \hat{\sigma}_1, \ldots, \hat{\sigma}_r)$, and then the refined embedding matrix is reconstructed as:

$$\hat{U} = \mathbf{U}\hat{\mathbf{\Sigma}}\mathbf{V}^T. \tag{7}$$

Finally, we use the first column of $\hat{U}$, denoted as $\hat{u}_a$, as the refined identity embedding to guide the identity preservation in the second stage.

### 3.3.2 Rotate-CLIP

Given a source face image $\mathbf{x}_a$ with source age $a$ and target age $b$, we extract the CLIP image embedding $i_a = I_{\mathrm{CLIP}}(\mathbf{x}_a)$, along with the text embeddings $t_a = T_{\mathrm{CLIP}}(a)$ and $t_b = T_{\mathrm{CLIP}}(b)$, using the pretrained CLIP image encoder $I_{\mathrm{CLIP}}(\cdot)$ and text encoder $T_{\mathrm{CLIP}}(\cdot)$. Our goal is to shift the age-related component in $i_a$ toward the target age domain in CLIP space, leveraging CLIP's joint visual-textual alignment. A common approach, inspired by [61], is to compute the age shift vector as the difference of text embeddings:

$$\Delta = t_b - t_a. \tag{8}$$

However, this simple subtraction may introduce semantic inconsistencies due to CLIP's coarse age representations [73, 77]. To address this, we propose a *rotational projection* using spherical linear interpolation (slerp), which more smoothly captures semantic transitions between ages:

$$\Delta' = \mathrm{slerp}(t_b, t_a, \lambda), \tag{9}$$

where $\lambda \in [0, 1]$ controls the interpolation. The Rotate-CLIP embedding is then defined as:

$$\hat{i}_a = i_a + \Delta', \tag{10}$$

which shifts $i_a$ toward the target age direction while preserving other identity-related information.

The refined identity embeddings $\hat{u}_a$ and $\hat{i}_a$, obtained from *SVR-ArcFace* and *Rotate-CLIP*, are projected through two MLPs to align with the text-embedding feature dimension, then concatenated to form *IDEmb* before injected into the cross-attention module of SDXL-Turbo:

$$\tilde{u}_a = \mathrm{MLP}_u(\hat{u}_a), \quad \tilde{i}_a = \mathrm{MLP}_i(\hat{i}_a). \tag{11}$$

### 3.4 Training Losses

Based on the architecture described above, we jointly optimize the MLPs and UNet-LoRA [23, 57] modules using a weighted combination of three objectives: identity loss, age loss, and quality loss. The ArcFace and CLIP encoders remain frozen during training.

**Identity Loss.** To preserve facial identity, we employ a combination of multi-scale structural similarity (MS-SSIM) [71] and high-level identity embedding similarity. Specifically, we use a pretrained ArcFace [12] model to extract embeddings and compute the cosine distance between the source image $\mathbf{x}_a$ and the generated image $\mathbf{x}_b$:

$$\mathcal{L}_{\mathrm{id}} = \lambda_1 \cdot (1 - \mathrm{MS\text{-}SSIM}(\mathbf{x}_a, \mathbf{x}_b)) + \lambda_2 \cdot (1 - \cos(f_{\mathrm{Arc}}(\mathbf{x}_a), f_{\mathrm{Arc}}(\mathbf{x}_b))), \tag{12}$$

where $f_{\mathrm{Arc}}(\cdot)$ denotes the ArcFace encoder.

**Age Loss.** To ensure age accuracy, we define an age loss that measures both visual consistency and numerical correctness. The first term computes the cosine similarity between embeddings of the intermediate result $\hat{\mathbf{x}}_b$ and the final output $\mathbf{x}_b$ using a pretrained MiVOLO [32] model. The second term minimizes the L2 distance between the predicted and target ages:

$$\mathcal{L}_{\mathrm{age}} = \lambda_3 \cdot (1 - \cos(f_{\mathrm{Mi}}(\hat{\mathbf{x}}_b), f_{\mathrm{Mi}}(\mathbf{x}_b))) + \lambda_4 \cdot \|g_{\mathrm{Mi}}(\mathbf{x}_b) - b\|_2^2, \tag{13}$$

where $f_{\mathrm{Mi}}(\cdot)$ and $g_{\mathrm{Mi}}(\cdot)$ denote the MiVOLO feature extractor and age estimator, respectively.

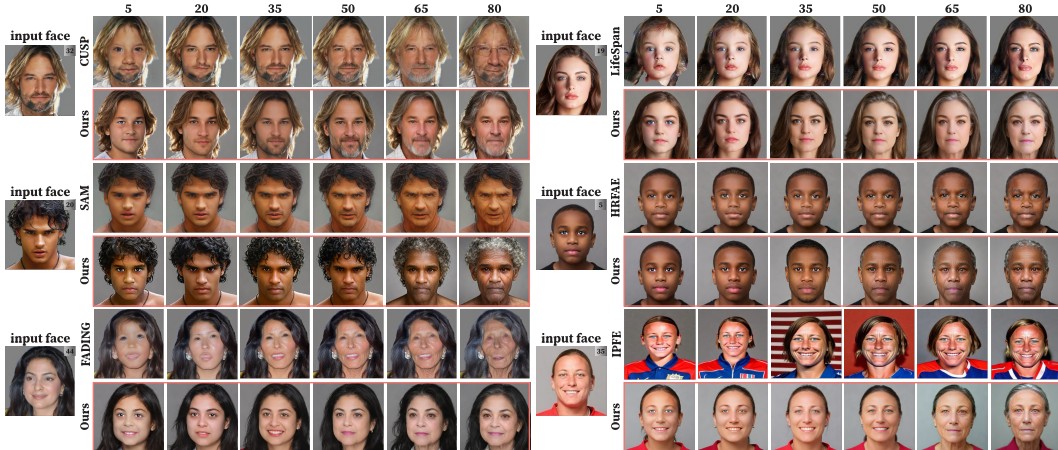

Figure 4: Qualitative comparison with existing face aging methods across lifespan ages. Our method *Cradle2Cane* is even able to imitate the natural hair change while the previous methods cannot. For comparisons on in-the-wild images, please refer to Fig. 8 in the Appendix.

**Quality Loss.** To improve perceptual fidelity, we combine the LPIPS metric [79], which measures perceptual similarity aligned with human vision, with an adversarial loss from a GAN [15] discriminator to encourage photorealism:

$$\mathcal{L}_{\text{per}} = \lambda_5 \cdot \text{LPIPS}(\mathbf{x}_a, \mathbf{x}_b) + \lambda_6 \cdot \mathcal{L}_{\text{GAN}}(\mathbf{x}_b), \tag{14}$$

**Overall Objective.** The final training objective is a weighted sum of the three losses:

$$\mathcal{L}_{\text{total}} = \mathcal{L}_{\text{id}} + \mathcal{L}_{\text{age}} + \mathcal{L}_{\text{per}}, \tag{15}$$

where $\lambda_1$ through $\lambda_6$ are scalar coefficients that balance the contributions of each component.

## 4 Experiments

### 4.1 Experimental Setups

**Evaluation Benchmarks and Metrics.** We evaluate our method on two datasets: a subset of CelebA-HQ [27] and CelebA-HQ (in-the-wild) test datasets. For each dataset, we randomly select 100 face images per gender. Each image is used to generate age-progressed faces from 0 to 80 years in 5-year intervals, resulting in 3,200 test images per dataset. We also use Carvekit [60] to remove background. Following prior works [14, 76], we utilize the Face++ API to quantitatively assess age estimation accuracy, identity preservation, and image quality. In addition, we employ large multimodal models, such as Qwen-VL [68], to conduct high-level perceptual evaluations. These models provide interpretable assessments of perceived age, identity consistency, and visual realism via carefully designed task-specific prompts. To jointly evaluate age accuracy and identity preservation, we propose the *Harmonic Consistency Score* (HCS), a unified metric that balances both factors. Full metric definitions and evaluation prompt templates are provided in Appendix A.

**Comparison Methods.** To evaluate the performance of our method, we compare it with several state-of-the-art face aging baselines. Specifically, we include: (1) Diffusion-based methods: IPFE [3], FADING [7]; and (2) GAN-based methods: SAM [1], CUSP [14], Lifespan [48], and HRFAE [76]. Detailed configurations and implementations of both our method and baselines are included in Appendix A.

### 4.2 Experimental Results

**Quantitative Comparison.** As demonstrated in Table 1, *Cradle2Cane* consistently outperforms existing face aging methods in both the Face++ and Qwen-VL evaluations across the CelebA-HQ and CelebA-HQ (in-the-wild) datasets. It achieves the lowest age estimation error, the highest image

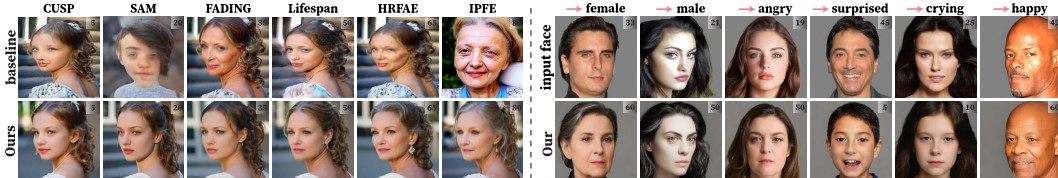

Figure 5: (Left) While applying to in-the-wild real human faces, *Cradle2Cane* demonstrates better performance while the existing methods often fail. (Right) Our *Cradle2Cane* can also be applied to modify gender and emotion attributes while performing age transformation on human faces.

Table 1: Quantitative comparison using both Face++ and Qwen-VL evaluation protocols on CelebA-HQ and CelebA-HQ (in-the-wild) test dataset. We calculate the age accuracy, identity preservation, image quality and the Harmonic consistency score (HCS) to compare with existing face aging methods. Best results are marked in  blue , and second-best in  green .

| Method | Type | Face++ Evaluation (CelebA-HQ) | | | | Qwen-VL Evaluation (CelebA-HQ) | | | | Qwen-VL (CelebA-HQ-in-the-wild) | | | | Inference Time (s) | Train Data |
|---|---|---|---|---|---|---|---|---|---|---|---|---|---|---|---|
| | | Age Diff. ↓ | ID Sim. ↑ | Img. Quality ↑ | HCS ↑ | Age Diff. ↓ | ID Sim. ↑ | Img. Quality ↑ | HCS ↑ | Age Diff. ↓ | ID Sim. ↑ | Img. Quality ↑ | HCS ↑ | | |
| Lifespan [48] | GAN | ±22.07 | 79.80 | 66.68 | 57.40 | ±27.99 | 71.99 | 86.03 | 42.38 | ±26.20 | 71.14 | 69.41 | 46.47 | 0.95 | 70K |
| HRFAE [76] | GAN | ±15.12 | 94.32 | 62.28 | 74.95 | ±17.77 | 77.86 | 90.93 | 62.19 | ±19.98 | 77.68 | 84.49 | 60.87 | 0.17 | 300K |
| SAM [1] | GAN | ±8.42 | 81.96 | 68.38 | 80.42 | ±6.31 | 72.15 | 90.70 | 77.72 | ±6.86 | 54.87 | 87.10 | 66.01 | 0.39 | 70K |
| CUSP [14] | GAN | ±9.59 | 85.92 | 64.98 | 80.67 | ±7.45 | 74.44 | 88.06 | 77.84 | ±13.66 | 76.89 | 81.86 | 70.94 | 0.24 | 30K |
| FADING [7] | Diffusion | ±14.47 | 86.70 | 64.65 | 73.52 | ±7.90 | 75.08 | 90.02 | 77.57 | ±9.25 | 73.33 | 88.01 | 75.06 | 61.26 | - |
| IPFE [3] | Diffusion | ±11.95 | 75.14 | 63.55 | 72.54 | ±11.67 | 69.40 | 87.01 | 70.01 | ±12.97 | 65.34 | 88.03 | 66.43 | 8.84 | - |
| *Cradle2Cane* | Diffusion | ±7.47 | 81.34 | 72.69 | 81.33 | ±4.62 | 70.29 | 92.37 | 78.33 | ±5.05 | 67.15 | 88.92 | 75.94 | 0.56 | 10K |

Table 2: Ablating each component with Qwen-VL evaluation.

| *AdaNI* | SVR-ArcFace | Rotate-Clip | Age Diff. ↓ | ID Sim. ↑ | Img. Quality ↑ | HCS ↑ |
|---|---|---|---|---|---|---|
| × | × | × | ±8.87 | 68.92 | 92.00 | 73.10 |
| ✓ | × | × | ±3.94 | 59.70 | 92.15 | 71.83 |
| × | ✓ | × | ±9.48 | 70.17 | 92.16 | 73.11 |
| ✓ | ✓ | × | ±6.75 | 63.38 | 92.43 | 71.92 |
| ✓ | ✓ | ✓ | ±4.62 | 70.29 | 92.37 | 78.33 |

Figure 6: User Study

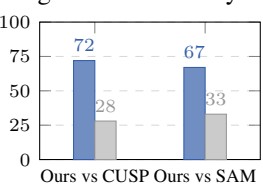

quality scores, and the best HCS values, while maintaining competitive identity preservation. Notably, *Cradle2Cane* achieves these results with a relatively small training set (10K) and a fast inference time (0.56s). These results underscore the effectiveness and efficiency of our framework in balancing aging realism, identity consistency, and visual quality across diverse evaluation protocols.

**Qualitative Comparison.** Figure 4 presents a visual comparison of face aging results between *Cradle2Cane* and recent GAN- and diffusion-based baselines. Compared to other methods, our approach demonstrates more realistic aging transitions with consistent identity preservation across all age ranges. In contrast, prior methods often exhibit texture artifacts, age realism issues, or identity shifts, particularly at extreme ages. Our method, however, produces natural skin aging, hair graying, and structural changes, reflecting a superior modeling of facial aging patterns. These results emphasize the visual fidelity and robustness of our framework.

**Ablation Study.** We conduct an ablation study to assess the impact of each proposed component, as shown in Table 2. Removing our aging mechanism *AdaNI* results in a substantial age estimation error (±8.87), emphasizing its critical role in achieving age accuracy. Introducing *AdaNI* alone significantly reduces the error (±3.94), though it slightly compromises identity similarity. Incorporating the SVR-ArcFace module improves identity consistency (from 59.70 to 63.38), validating its effectiveness for identity preservation. Finally, adding Rotate-Clip further enhances identity performance and contributes to a well-balanced trade-off across all metrics. Notably, the overall HCS score steadily increases throughout, with the full configuration achieving the highest score (78.33).

**User Study.** To assess human-perceived quality, we conduct a user study comparing our method *Cradle2Cane* with two state-of-the-art face aging methods: SAM [1] and CUSP [14]. We randomly sample 20 identity images from the CelebA-HQ test set and generate 6 aging results for each, evenly spaced from age 5 to 80. 50 volunteers are asked to perform pairwise comparisons between our results and each baseline, considering three joint criteria—age accuracy, identity preservation, and overall image quality. Each query follows a forced 1-vs-1 protocol with randomized display order to prevent position bias. As summarized in Fig. 6, our method is consistently preferred by a clear majority, demonstrating superior perceptual quality and better alignment with human judgment.

**Additional Applications.** Since our method *Cradle2Cane* is build upon the large T2I diffusion model, it is also able to deal with various in-the-wild images while the previous methods fail (Fig. 5-(Left). Besides facial age transformation, our method can be easily adapted to other facial editing tasks, such as gender transformation and expression modification. As illustrated in Fig. 5-(Right), our approach achieves gender and expression changes while maintaining high identity consistency. This further demonstrates the versatility and generalizability of our facial editing framework.

# 5  Conclusion

In this work, we tackle the fundamental challenge of achieving both age accuracy and robust identity preservation in face aging—a problem we term the *Age-ID trade-off*. While existing methods often prioritize one objective at the expense of the other, our proposed framework, *Cradle2Cane*, introduces a two-pass framework that explicitly decouples these goals. By leveraging the flexibility of few-step text-to-image diffusion models, we introduce an adaptive noise injection (*AdaNI*) mechanism for fine-grained age control in the first pass, and reinforce identity consistency through dual identity-aware embeddings (*IDEmb*) in the second pass. Our method is trained end-to-end, enabling high-fidelity, controllable age transformation across the full lifespan, while significantly improving inference speed and visual realism. Extensive evaluations on CelebA-HQ confirm that *Cradle2Cane* achieves new state-of-the-art performance in terms of both age accuracy and identity preservation. In addition, *Cradle2Cane* demonstrates strong generalization to real-world scenarios by effectively handling in-the-wild human face images, a setting where existing methods often fail.

# 6  Limitations and Boarder Impacts

**Limitations**  While our method achieves state-of-the-art performance in balancing age realism and identity consistency, there remain several limitations that merit discussion. In cases of extreme age transformation (e.g., from a child to an elderly person or vice versa), the model tends to favor facial realism and age accuracy at the cost of preserving some visual details in the original image. For example, accessories such as eyeglasses, earrings, or clothing color may not always be faithfully retained after editing, as they are not explicitly modeled or enforced during training. This issue stems from the adaptive noise injection design, which purposefully increases editability for large age gaps, potentially altering finer image semantics beyond facial identity.

**Broader Impacts**  Our proposed face aging method provides a flexible framework for independently controlling visual age via a two-pass diffusion process. This enables a range of positive applications, including digital entertainment (e.g., age effects in movies or games), age-invariant face recognition, and future appearance simulation for healthcare or counseling. All experiments are conducted on anonymized public benchmarks under ethical research settings.

At the same time, we recognize the potential risks associated with misuse. The ability to generate photorealistic age manipulation with identity consistency may facilitate malicious uses such as identity spoofing, misinformation, or privacy violations. We strongly discourage unauthorized or commercial deployment without safeguards like watermarking, traceable provenance, or human review. We hope this work inspires further progress toward ethical and responsible generative modeling in the vision community.

# Acknowledgements

This research was supported by the collaborative project between Beijing Samsung Telecommunications Technology Co., Ltd. and the Tianjin Key Laboratory of Visual Computing and Intelligent Perception (VCIP) of Nankai University, entitled "Generated Face for Enhancing Face Dataset" (Project No. SRC-Beijing-DVL-2024-00241).

We would like to express our sincere gratitude to all co-authors for their invaluable contributions and insightful suggestions. We are particularly grateful to Yaxing Wang (Associate Professor, Nankai University) and Kai Wang (Assistant Professor, City University of Hong Kong (Dongguan)). Their meticulous advice and guidance were instrumental in the completion of this research.

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

# Appendix

## A    Experiments details

### A.1    Implementation Details.

Our framework is built upon the SDXL-Turbo architecture and trained on the FFHQ dataset, which contains high-quality facial images annotated with age and gender labels. To improve background consistency, we employ CarveKit [60] for foreground-background segmentation, replacing non-facial regions with a uniform gray mask. During training, we guide the model using text prompts in the format: *"a face image of a {target age} years old {gender}"* where *gender* is either *female* or *male*. All images are generated at a fixed resolution of 512×512.

The U-Net backbone of SDXL-Turbo is fine-tuned via LoRA, and training is conducted on 8 NVIDIA A6000 GPUs with a batch size of 4. For hyperparameters, in *SVR-ArcFace*, we set $\alpha = 0.01$ and $\beta = 1.2$, while in *Rotate-CLIP*, we use an interpolation strength of $\lambda = 0.5$. For the training loss terms, we set the weights as $\lambda_1 = 0.25$, $\lambda_2 = 1.2$, $\lambda_3 = 1.5$, $\lambda_4 = 1.5$, $\lambda_5 = 0.25$, and $\lambda_6 = 0.1$.

### A.2    Prompts for Qwen-VL Evaluation

To further evaluate the performance of our method, we leverage Qwen-VL [68] for perceptual evaluation across three key aspects: age accuracy, identity consistency, and image quality. The prompts are structured as follows:

**Age Estimation**
> *"Please detect the age of the person in the image and return in the following format: age:{age}."*

**Image Quality**
> *"Please evaluate image quality of the face image and provide a quality score (0–100), and return in the following format: quality:{quality_score}."*

**Identity Similarity**
> *"Please evaluate if the individuals in these two images are the same person based solely on facial structure, ignoring factors such as style, lighting, age, or background. Provide a score between 1 (completely different) and 100 (completely identical), and return in the following format: similarity:{similarity_score}."*

Each generated image is assessed using the corresponding prompt. When calculate identity similarity, the reference image is presented alongside the generated aged image to facilitate comparison. The resulting textual responses from Qwen-VL are parsed to extract quantitative scores.

### A.3    Harmonic Consistency Score (HCS)

To jointly evaluate age accuracy and identity similarity, we introduce the *Harmonic Consistency Score* (HCS), defined as:

$$A = \left( 1 - \frac{\text{MAE}}{M} \right) \cdot 100, \quad \text{HCS} = 2 \cdot \frac{A \cdot I}{A + I},$$

where MAE denotes the mean absolute error between the predicted and target ages, and $M$ is the predefined maximum allowable age deviation (set to 40). The normalized age accuracy $A \in [0, 100]$ reflects proximity to the target age, while $I \in [0, 100]$ is the identity similarity score, obtained by multiplying the cosine similarity between ArcFace embeddings by 100. The harmonic formulation ensures a balanced evaluation that penalizes degradation in either attribute. Compared to simple averaging, the harmonic mean is more sensitive to low values, which is desirable in this context: a high HCS can only be achieved when both age accuracy and identity similarity are simultaneously high.

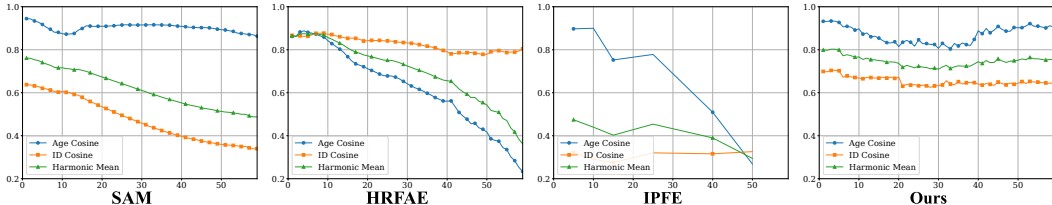

Figure 7: *Age–ID trade-off* curves of different methods. As the age shift value increases, either the Age cosine or ID cosine decreases for SAM, HRFAE, and IPFE. In contrast, our method maintains stable Age and ID consistency, showing no significant drop.

## A.4 Open-Source Implementations and Settings

For reproducibility and comprehensive comparison, we evaluate several open-source face aging methods using their official pretrained models and inference pipelines. The following repositories are utilized:

| Method | Repository Link |
|---|---|
| SAM [1] | https://github.com/yuval-alaluf/SAM |
| IPFE [3] | https://github.com/sudban3089/ID-Preserving-Facial-Aging |
| FADING [7] | https://github.com/MunchkinChen/FADING |
| CUSP [14] | https://github.com/guillermogotre/CUSP |
| Lifespan [48] | https://github.com/royorel/Lifespan_Age_Transformation_Synthesis |
| HRFAE [76] | https://github.com/InterDigitalInc/HRFAE |

Table 3: Open-source face aging methods and their official repositories.

All models are evaluated using standardized input settings and tested on the CelebA-HQ and CelebA-HQ (in-the-wild) test dataset. Due to the licensing terms of the CelebA [41] dataset , we are unable to display the original input images. Instead, we present the corresponding image inversions generated using null-text inversion [45]. We report metrics including age accuracy, identity similarity, image quality, and the proposed HCS. This unified evaluation protocol ensures fair and consistent performance comparison across diverse methods. For IPFE, which requires multiple images of the same identity as input, we randomly select one reference image per subject for identity similarity evaluation. Since both IPFE and FADING perform test-time tuning for each new input face image, a fixed training dataset is not applicable to these methods, and thus their training data size is not reported.

## A.5 Age–ID Trade-off Evaluation Details

To quantitatively evaluate the trade-off between age accuracy and identity consistency, we selected 50 male and 50 female face images from the CelebA-HQ test dataset. For each image, we generated aging results across age offsets ranging from $-60$ to $+60$ with a step size of $1$ year, excluding the zero offset. Identity similarity was measured using ArcFace by computing the cosine similarity between the original and age-edited images. Age similarity was quantified based on the predicted age error from the MiVOLO estimator. Specifically, we computed the age consistency score as $\text{age\_cosine} = 1 - \frac{|\text{age}_{\text{pred}} - \text{age}_{\text{target}}|}{\text{max\_age\_diff}}$, where max_age_diff is set to $40$. This score ranges from $0$ to $1$, with higher values indicating better age alignment. Comparisons with the remaining methods are illustrated in Fig. 7. Note that IPFE [3] does not support continuous year-level age control, thus only the provided age groups reported in the original paper were included in our evaluation.

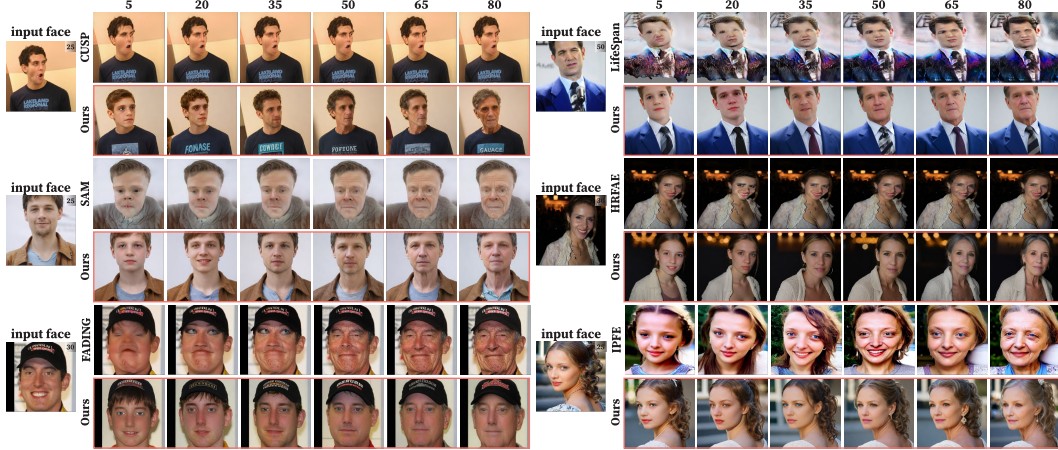

Figure 8: Qualitative comparison of aging results on CelebA-HQ (in-the-wild) images. Despite the challenges posed by real-world conditions such as occlusions, varying poses, and complex lighting and backgrounds, our method generates more photorealistic and coherent aging results, with better preservation of identity and more accurate aging effects including wrinkle formation, hair graying, and facial structure changes.

## A.6  User Study Details

We conducted a user study to evaluate the perceptual quality of age-transformed face images generated by our method in comparison to two state-of-the-art methods, CUSP and SAM. A total of 50 volunteers participated in the survey. We randomly selected 20 identities from the CelebA-HQ test dataset and generated 6 age-progressed images for each, evenly spaced across ages ranging from 5 to 80 years. In each trial, participants were presented with pairs of image groups—each group consisting of the original reference face followed by the corresponding age-transformed images. The two groups (our method versus another method) were displayed side-by-side with randomized order to mitigate positional bias.

As illustrated in Fig. 12, the questionnaire provided clear guidance instructing participants to jointly assess three criteria: age realism, identity similarity, and overall visual quality. An example with labeled groups and comparative explanations was included to familiarize participants with the evaluation process. The study employed a forced-choice 1-vs-1 protocol, and results summarized in the main text demonstrate a consistent preference for our approach, confirming its superior ability to generate visually convincing and identity-preserving age transformations.

## B  Additional Results

### B.1  Qualitative Comparison on in-the-wild Images

Fig. 8 presents a qualitative comparison of aging results on CelebA-HQ (in-the-wild) images, evaluating our method against state-of-the-art GAN-based and diffusion-based approaches. Compared to aligned datasets, in-the-wild images pose greater challenges due to diverse facial poses, complex backgrounds, occlusions, and uncontrolled lighting conditions. Under these challenging scenarios, the baseline methods exhibit various limitations. While CUSP and HRFAE maintain relatively high identity consistency, they often fail to capture realistic aging cues, resulting in over-smoothed faces with insufficient detail such as wrinkles and hair graying. LifeSpan, SAM, and Fading are prone to producing severe artifacts and significant identity drift, particularly under large age transformations, leading to unnatural facial structures and distorted textures. IPFE, on the other hand, generates faces with low visual fidelity and suffers from notable identity inconsistency, often producing blurry or distorted outputs. In contrast, our method demonstrates strong robustness and generalization in these real-world conditions, consistently generating high-fidelity aging results that preserve identity features while capturing fine-grained and realistic age-related changes such as wrinkle formation, hair graying, and structural facial transitions.

## B.2  Generalization to Diverse Reference Ages

To thoroughly evaluate the age controllability and robustness of our method, we conduct face aging experiments using a wide range of reference ages, spanning from 1 to 80 years old. Specifically, we select reference images at 10-year intervals and generate aging results targeting six representative ages: 5, 20, 35, 50, 65, and 80, for each reference image. As shown in Fig. 11, our model demonstrates smooth and realistic age transformations across the entire age range, effectively handling both forward and backward aging transitions. The generated results exhibit consistent aging patterns, such as the gradual appearance of wrinkles, changes in skin texture, facial structural modifications, and hair graying, while preserving identity fidelity at each age target. These results highlight the strong generalization ability of our approach, ensuring effective age transformation across a variety of reference faces with diverse age inputs.

## B.3  Face Aging across the Entire Lifespan

To further evaluate the age controllability of our approach, we conduct experiments generating human faces across the full age spectrum from 1 to 80 years. As shown in Fig. 9 and Fig. 10, our model produces smooth and continuous transitions of facial features across decades, accurately reflecting both age progression and regression. In contrast to prior works [3, 14, 76], which are typically limited to coarse age intervals (e.g., child, adult, elderly) or restricted age ranges, our method supports fine-grained age conditioning at each individual year without the need for additional retraining or manual tuning.

# C  Additional Experiment

## C.1  Additional Quantitative Experiment

To further validate the effectiveness of our approach, we conducted additional quantitative experiments on extra datasets and baselines. Specifically, we compared our method with recent generic face-editing systems (e.g., StyleCLIP [50] and FaceDNeRF [78]) on the standard aging datasets AgeDB-30 [46], CACD [6], and FG-NET. For each dataset, we generated approximately 200 images per method and evaluated them using four metrics: age difference (Age Diff.), identity similarity (ID Sim.), image quality (Img. Quality), and Holistic Consistency Scores (HCS). The comprehensive results are summarized in Table 4, where lower Age Diff. and higher scores on the remaining metrics indicate better performance.

| Method | AgeDB-30 | | | | CACD | | | | FG-NET | | | |
|---|---|---|---|---|---|---|---|---|---|---|---|---|
| | Age Diff. ↓ | ID Sim. ↑ | Img. Quality ↑ | HCS ↑ | Age Diff. ↓ | ID Sim. ↑ | Img. Quality ↑ | HCS ↑ | Age Diff. ↓ | ID Sim. ↑ | Img. Quality ↑ | HCS ↑ |
| CUSP [14] | 8.29 | 76.30 | 82.80 | 77.75 | 10.80 | 75.40 | 85.86 | 74.18 | 19.45 | 74.20 | 79.79 | 60.71 |
| SAM [1] | 8.72 | 58.91 | 86.57 | 67.19 | 9.25 | 67.27 | 87.57 | 71.75 | 5.33 | 72.18 | 81.98 | 78.77 |
| FADING [7] | 8.10 | 76.00 | 79.22 | 77.82 | 5.16 | 71.27 | 86.44 | 78.39 | 7.22 | 73.15 | 79.43 | 77.30 |
| Styleclip [50] | 14.73 | 62.12 | 86.60 | 62.64 | 16.28 | 70.64 | 87.80 | 64.48 | 14.73 | 73.82 | 88.19 | 68.08 |
| Facednerf [78] | 12.60 | 53.78 | 91.97 | 60.25 | 13.05 | 59.00 | 92.02 | 62.91 | 13.21 | 56.37 | 92.43 | 61.21 |
| *Cradle2Cane* | 5.57 | 73.00 | 90.01 | 79.00 | 4.63 | 66.73 | 90.02 | 76.06 | 5.79 | 67.18 | 85.03 | 75.25 |

Table 4: Comparison on AgeDB-30, CACD and FG-NET datasets. Best results are marked in blue , and second-best in green .

Our method, *Cradle2Cane*, demonstrates exceptional performance across all three benchmarks. On AgeDB-30, it achieves state-of-the-art results, securing the best age accuracy and the top harmonic score, which underscores its superior overall balance. This strong performance continues on the CACD dataset, where our method again delivers the best age accuracy while maintaining highly competitive scores on other metrics. Even on the highly challenging FG-NET dataset, *Cradle2Cane* proves its robustness by delivering the second-best age accuracy and demonstrating strong, consistent performance against all competitors.

## C.2  Ablation Study on Threshold Selection

To investigate the rationale behind our choice of age thresholds, we conducted an ablation study with multiple division settings. Specifically, we compared thresholds of {5,15}, {5,20}, {7,22}, {10,30}, {15,35}, and {20,40}. The results presented in Table 5 reveal a distinct trade-off between

age accuracy and identity preservation. We observe that wider threshold intervals, such as {15,35} and {20,40}, yield superior identity similarity (ID Sim.) and Holistic Consistency Scores (HCS). However, this gain is achieved at the expense of age fidelity, as evidenced by the significant increase in the Age Difference metric from a low of 4.92 to 6.71.

Given that age accuracy is the central objective of our work, we selected the {5, 20}division as our default configuration. This setting achieves the best performance in age accuracy while maintaining a competitive balance in identity preservation and image quality. This choice ensures our primary goal is met and establishes a rigorous, transparent baseline for our experiments.

Table 5: Ablation study on different threshold divisions. This analysis highlights the trade-off between age accuracy and identity preservation. Best results are marked in  blue , and second-best in  green .

| Thresholds | Age Diff. ↓ | ID Sim. ↑ | Img. Quality ↑ | HCS ↑ |
|---|---|---|---|---|
| {5, 15} | 4.93 | 73.18 | 92.28 | 79.77 |
| {5, 20} | 4.92 | 73.73 | 92.60 | 80.11 |
| {7, 22} | 5.06 | 74.82 | 92.89 | 80.60 |
| {10, 30} | 5.46 | 76.91 | 92.56 | 81.36 |
| {15, 35} | 5.64 | 77.09 | 92.82 | 81.26 |
| {20, 40} | 6.71 | 78.55 | 92.74 | 80.82 |

## C.3 Ablation Study on Robustness and Architectural Contributions

To verify whether the robustness of our method on in-the-wild face images primarily comes from background removal pre-processing (Carvekit) or from the model architecture itself, we conducted an ablation study. Specifically, we compared our method with and without Carvekit pre-processing. Both variants were trained for 10 epochs on 1,000 images and evaluated on the CelebA-in-the-wild dataset. The results are shown in Table 6.

Table 6: Ablation study of the background removal pre-processing (Carvekit). The minor performance difference highlights that robustness is intrinsic to the model architecture. Best results are marked in  blue .

| Method | Age Diff. ↓ | ID Sim. ↑ | Img. Quality ↑ | HCS ↑ |
|---|---|---|---|---|
| w/ Carvekit | 6.76 | 62.00 | 91.75 | 71.02 |
| w/o Carvekit | 7.00 | 62.27 | 91.60 | 70.97 |

As presented in the table, the performance impact of background removal is marginal. This finding indicates that Carvekit serves as a beneficial but non-essential preprocessing step, rather than the primary source of the model's robustness. Instead, the method's resilience to in-the-wild variations is primarily attributed to its architectural design. First, the SDXL-Turbo backbone, pre-trained on a vast and diverse dataset, provides a strong foundation for generalization across varied poses, lighting conditions, and expressions. Second, our proposed IDEmb module systematically reinforces identity preservation. The SVR-ArcFace component extracts a stable identity embedding that is disentangled from transient attributes, while Rotate-CLIP executes minimal, precise modifications within CLIP's robust semantic space. This dual mechanism ensures that the age attribute is altered while other original characteristics are preserved with high fidelity. In summary, this study confirms that our model's robustness is an intrinsic property of its architecture, with SDXL-Turbo enabling generalization and IDEmb ensuring identity-consistent editing.

**Algorithm 1** The Proposed *Cradle2Cane* Framework

---

1: **Input:** Source image $\mathbf{x}_a$, source age $a$, target age $b$.
2: **Output:** Final aged image $\mathbf{x}_b$.

3: **// Pass 1: Adaptive Age Transformation**
4: Select noise level $\mathbf{z}_i$ based on $|\Delta\text{age}|$ per (Eq. 1).
5: $c_{\text{age}} \leftarrow \text{CLIP\_Encoder}(\text{age prompt for } b)$
6: Denoise from noise level $\mathbf{z}_i$ with condition $c_{\text{age}}$ to get latent $\hat{\mathbf{z}}_0$.
7: $\hat{\mathbf{x}}_b \leftarrow D(\hat{\mathbf{z}}_0)$ ▷ $D$ is VAE Decoder; $\hat{\mathbf{x}}_b$ is the intermediate image

8: **// Pass 2: Identity Enhancement**
9: **// — *IDEmb* Generation —**
10: Generate aged variants $\{\mathbf{x}_b^{(i)}\}_{i=1}^n$ from $\mathbf{x}_a$.
11: $u_a \leftarrow \text{ArcFace}(\mathbf{x}_a)$, $\{u_b^{(i)}\} \leftarrow \text{ArcFace}(\{\mathbf{x}_b^{(i)}\})$
12: $U \leftarrow [u_a, u_b^{(1)}, \ldots, u_b^{(n)}]$ ▷ (Eq. 2)
13: $\mathbf{U}, \mathbf{\Sigma}, \mathbf{V}^T \leftarrow \text{SVD}(U)$ ▷ (Eq. 3)
14: $\hat{\mathbf{\Sigma}} \leftarrow \text{Reweight}(\mathbf{\Sigma})$ with (Eq. 4).
15: $\hat{U} \leftarrow \mathbf{U}\hat{\mathbf{\Sigma}}\mathbf{V}^T$ ▷ (Eq. 5)
16: $\hat{u}_a \leftarrow \hat{U}[:, 0]$ ▷ Refined SVR-ArcFace embedding
17:
18: $i_a \leftarrow I_{\text{CLIP}}(\mathbf{x}_a)$; $t_a \leftarrow T_{\text{CLIP}}(a)$; $t_b \leftarrow T_{\text{CLIP}}(b)$
19: $\Delta' \leftarrow \text{slerp}(t_b, t_a, \lambda)$ ▷ (Eq. 7)
20: $\hat{i}_a \leftarrow i_a + \Delta'$ ▷ Refined Rotate-CLIP embedding (Eq. 8)
21:
22: **// — Final Refinement —**
23: $\tilde{u}_a \leftarrow \text{MLP}_u(\hat{u}_a)$; $\tilde{i}_a \leftarrow \text{MLP}_i(\hat{i}_a)$ ▷ (Eq. 9)
24: $c_{\text{ID}} \leftarrow \text{concat}(\tilde{u}_a, \tilde{i}_a)$ ▷ Form *IDEmb*
25: Set low noise timestep $T_{\text{low}}$.
26: $\mathbf{z}_{T_{\text{low}}} \leftarrow \text{AddNoise}(E(\hat{\mathbf{x}}_b), T_{\text{low}})$ ▷ $E$ is VAE Encoder
27: **for** $t = T_{\text{low}}, \ldots, 1$ **do** $\mathbf{z}_{t-1} \leftarrow p_\theta(\mathbf{z}_t, t, c_{\text{ID}})$ ▷ Denoise with identity condition
28: **end for**
29: $\mathbf{x}_b \leftarrow D(\mathbf{z}_0)$ ▷ Get the final result
30:
31: **Return** $\mathbf{x}_b$.

---

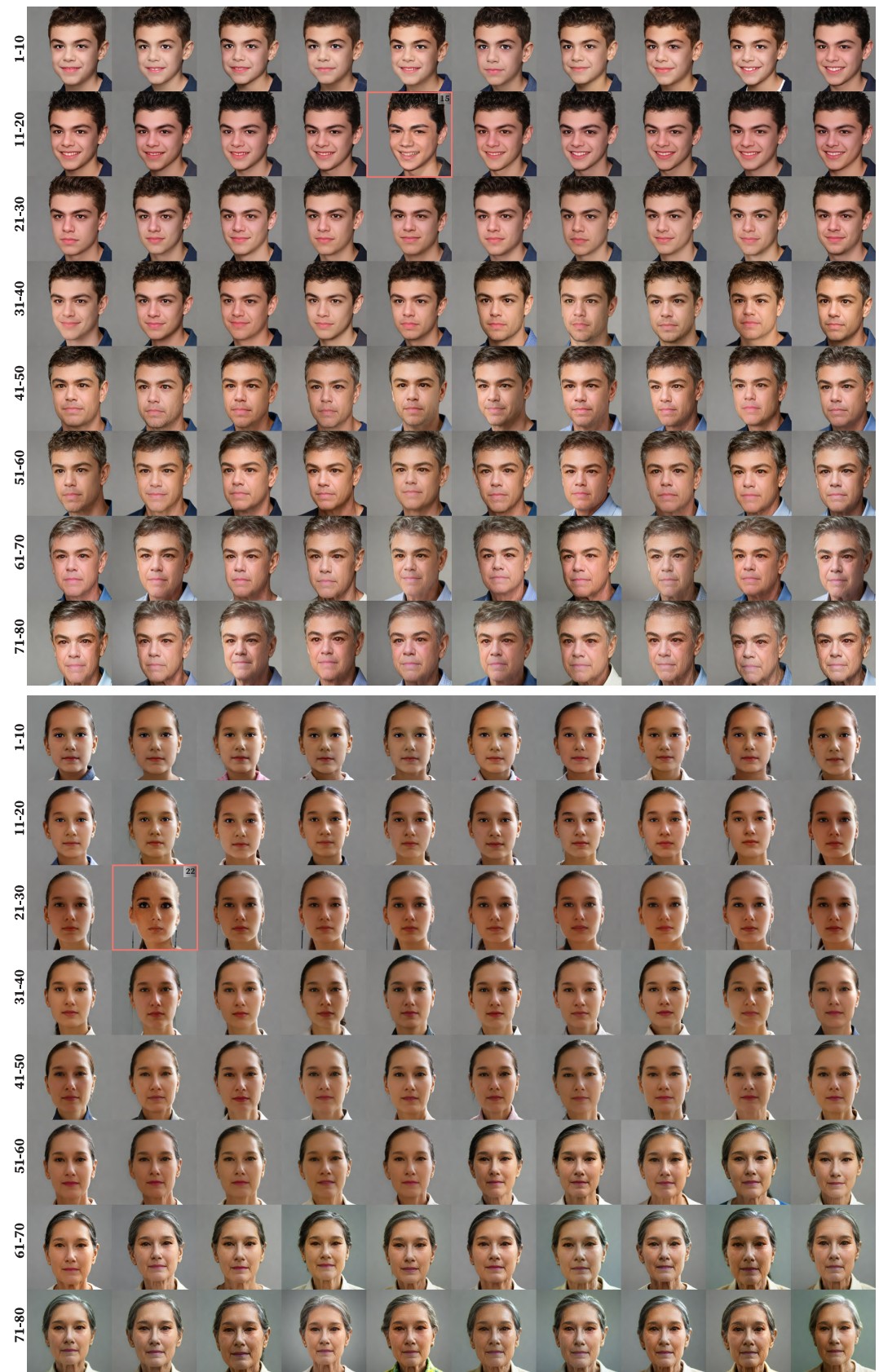

Figure 9: Face aging results from 1 to 80 years old. Reference images are marked in red.

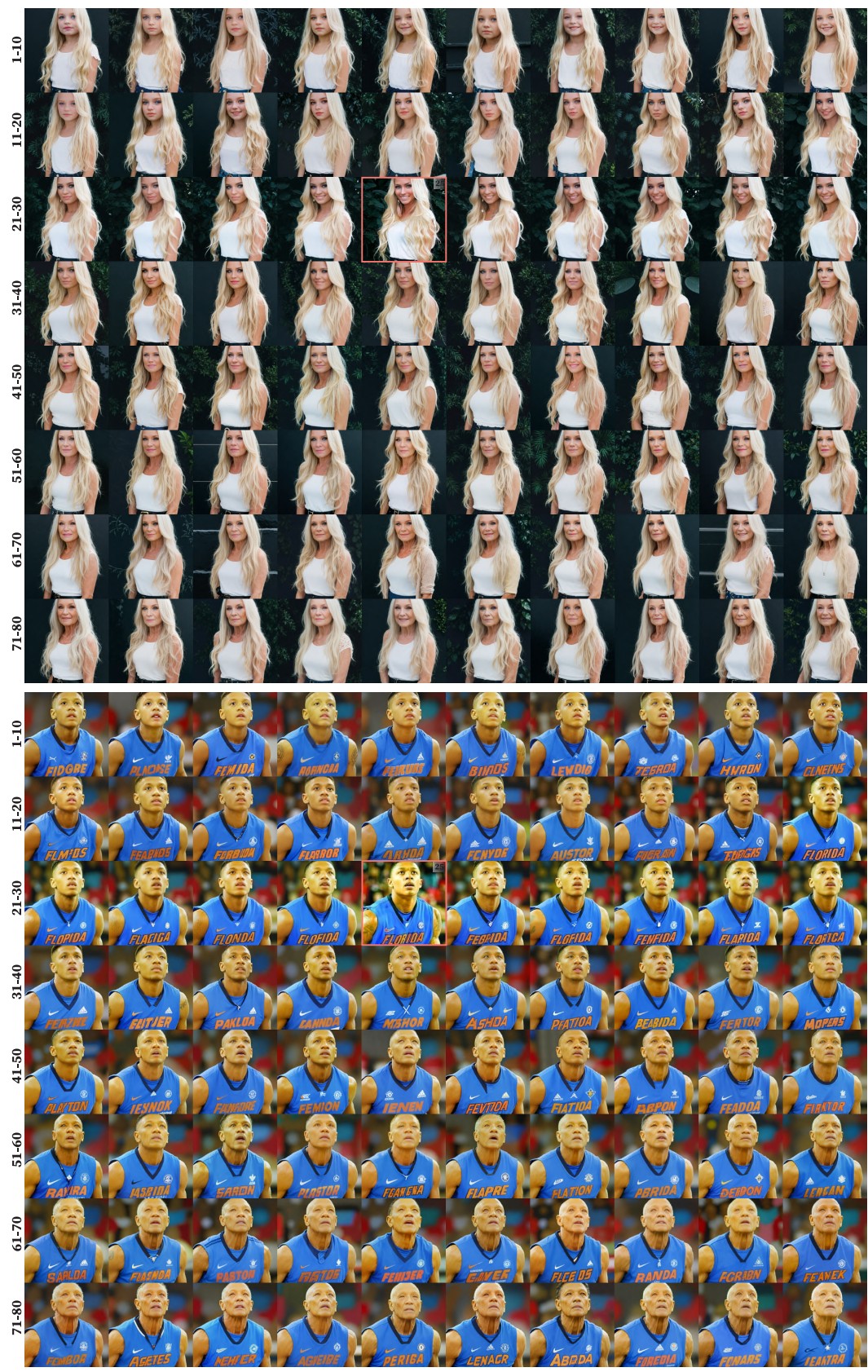

Figure 10: Face aging results from 1 to 80 years old of in-the-wild images.

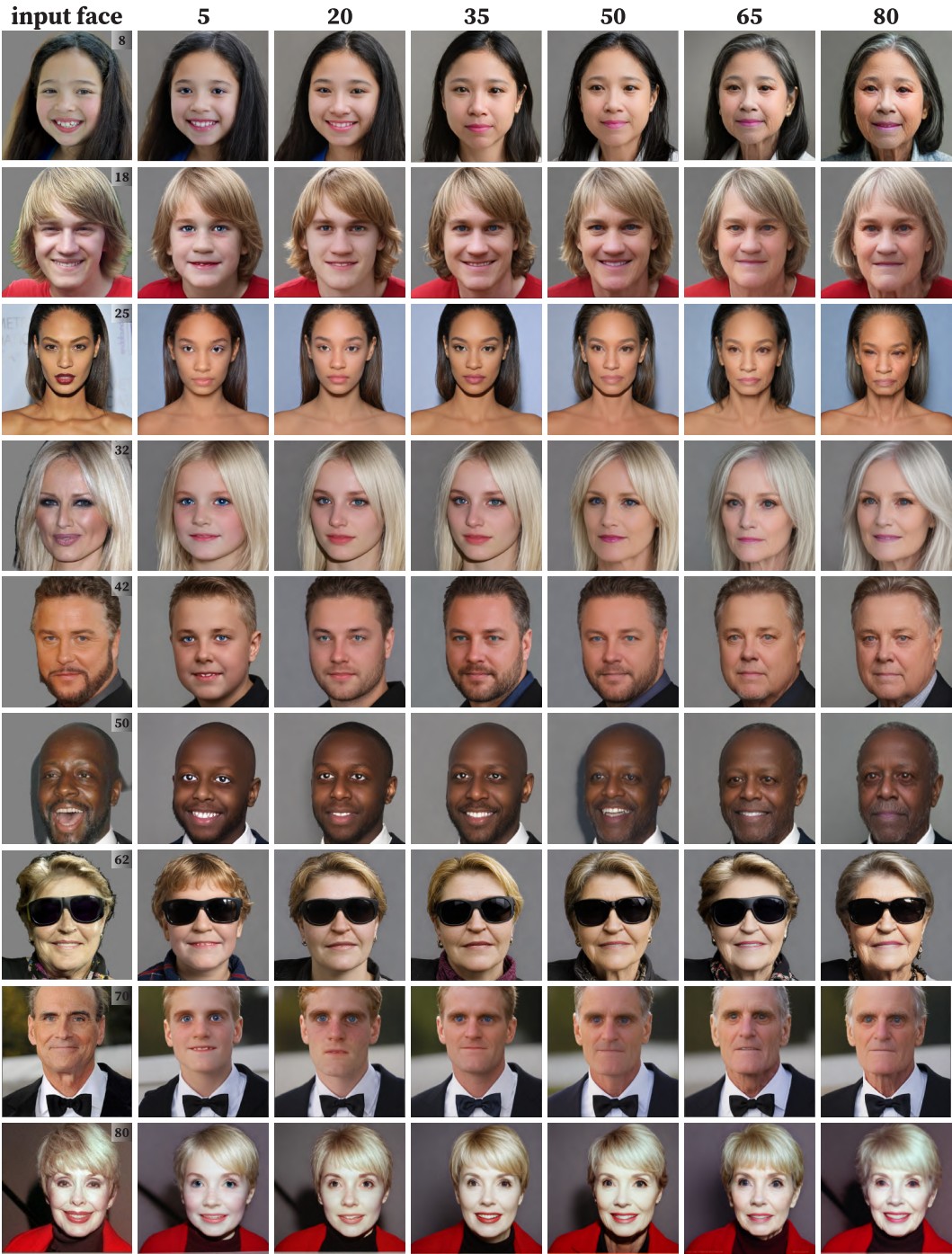

Figure 11: Aging results generated from diverse reference ages. For each reference image, we synthesize faces at six target ages: 5, 20, 35, 50, 65, and 80. Our method produces smooth and realistic age transitions across the entire lifespan, capturing both forward and backward aging effects while maintaining high identity consistency.

# Facial Age Transformation Quality Survey

## 📝 Instructions

In the following questionnaire, you will be shown a series of paired image sets (an original face image and its corresponding age-transformed versions). We kindly ask you to subjectively assess the quality of the generated images based on the three dimensions below:

## ⚖️ Evaluation Criteria

### 1. Age Realism

Does the age progression or regression in the generated images resemble natural human aging?
For example:
- Do elderly faces have realistic wrinkles or gray hair?
- Do children's faces look appropriately young and smooth?

### 2. Identity Similarity

Does the person in the transformed image still look like the original individual?
Focus on:
- Consistency in facial structure, shape, and distinctive features.
- Whether the identity remains recognizable across age changes.

### 3. Visual Realism & Quality

Do the generated images look natural and visually appealing?
Consider:
- Presence of artifacts, blurriness, or distortions.
- Overall consistency and image clarity.

## 📷 Image Comparison Task

In each question, you will see two groups of results labeled A and B.
Each group includes:
- One reference image (original face)
- Six generated images, representing ages from 5 to 80 years old

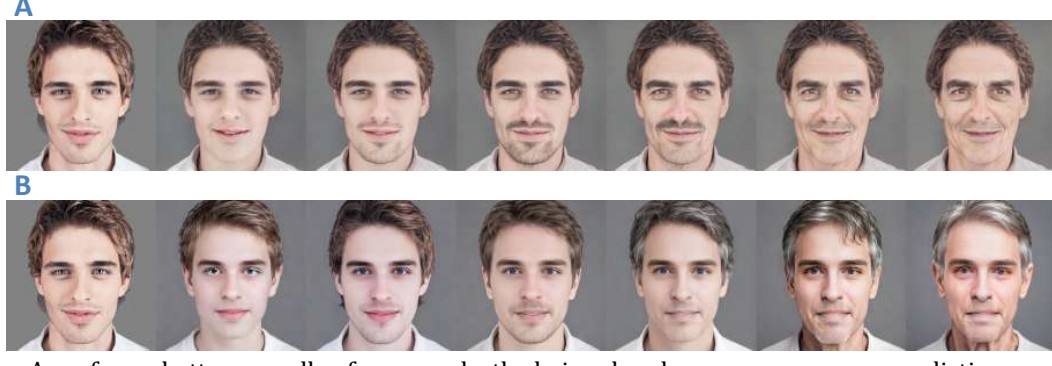

A performs better overall— for example, the hair color changes appear more realistic
, and the image quality is better.

Figure 12: User study setup for comparing the visual outcomes of age-transformed face images. Participants were presented with pairs of image groups, each showing the original face alongside age-transformed images from our method and the baselines (CUSP and SAM). They assessed three criteria: age realism, identity similarity, and overall visual quality.

