# OpenReview forum: "From Cradle to Cane: A Two-Pass Framework for High-Fidelity Lifespan Face Aging"
_NeurIPS.cc/2025/Conference — NeurIPS 2025 poster_

### Official Review · Reviewer_kgYz · 2025-06-26

**Clarity:** 3
**Significance:** 2
**Originality:** 2
**Rating:** 4
**Confidence:** 3

**Summary:**

This paper proposes a two-stage high-fidelity facial aging framework named Cradle2Cane, built upon a few-step text-to-image diffusion model. The first stage employs adaptive noise injection, dynamically adjusting the noise level using age and gender prompts. The second stage introduces an identity-aware embedding module (IDEmb), which integrates SVR-ArcFace and Rotate-CLIP to enhance identity preservation, enabling end-to-end training. Experiments demonstrate that the proposed framework outperforms existing methods on the CelebA-HQ dataset in terms of age accuracy, identity consistency, and image quality. The core concern of this paper is about its novelty and results.

**Questions:**

please refer to the weakness part

**Ethical Concerns:**

["NO or VERY MINOR ethics concerns only"]

**Final Justification:**

Thanks for the rebuttal. I keep my original Borderline accept

**Limitations:**

yes

**Quality:**

3

**Strengths And Weaknesses:**

Strengths:
1. The paper is well-written, and the experimental setup is solid. The key contributions are well-validated through experiments.
2. The approach to extract an identity-aware and age-independent representation is reasonable, leveraging SVD decomposition and CLIP-space interpolation.

Weaknesses:
1. The novelty of the experimental components is relatively limited. Techniques like SVD decomposition and CLIP-space interpolation have been extensively explored during the StyleGAN era. The first stage is essentially an image-to-image translation (SD-edit).
2. The results indicate that some age-irrelevant features may still be altered. For instance, in Fig. 9, the male subject's clothing changes, which is not expected to correlate with aging.

---

> ### Author Rebuttal · Authors · 2025-07-31
>
> We sincerely appreciate your constructive feedback. We are grateful for your recognition that the paper is ***well-written*** and the experiments are ***solid***; that our proposed framework is ***reasonable*** and ***well-validated***. Below, we provide detailed responses to each of your specific points.
>
> **W1: The novelty of the experimental components is relatively limited. Techniques like SVD decomposition and CLIP-space interpolation have been extensively explored during the StyleGAN era. The first stage is essentially an image-to-image translation (SD-edit).**
>
>
> We agree and acknowledge that the individual components we employ, such as SVD reweighting, and CLIP/ArcFace embeddings, have established precedents in the broader fields of generative modeling and diffusion-based editing.
>
> However, we respectfully argue that our core novelty lies not in the invention of the base components, but in the creation of a novel framework to solve a challenging and previously under-explored problem: the persistent Age-Identity (Age-ID) tradeoff in facial age editing.
>
> Our main contributions and points of novelty are:
>
> ***(1) Problem Formulation and Analysis***: To the best of our knowledge, *we are the first to conduct an in-depth, systematic investigation of the Age-ID tradeoff*. While implicitly known, this problem has not been formally analyzed or addressed with a targeted solution. Our work formally identifies this critical challenge, substantiating it with empirical evidence (Fig. 1 and Fig. 2), which itself is a valuable contribution to the field.
>
> ***(2) Novel Two-Pass Architectural Design***: Our central innovation is Cradle2Cane, a novel two-pass framework architected specifically to resolve the Age-ID tradeoff. *Prior methods attempt to modify age and preserve identity simultaneously in a single, entangled step, leading to suboptimal results. Our approach's novelty lies in decoupling these competing objectives*. By separating age accuracy from identity preservation into two distinct passes, we introduce a new and effective strategy that fundamentally differs from previous single-pass editing paradigms.
>
> ***(3) Fine-Grained Control via Adaptive Noise Injection***: While the concept of noise controlling editing strength is known, our approach introduces an adapative mechanism. We propose an adaptive noise injection strategy that calibrates noise intensity based on the magnitude of the desired age transformation. Specifically, our method dynamically adjusts the degree of injected noise for each edit—a small age change receives a subtle noise injection, while a large transformation receives a proportionally stronger one. *This marks a crucial departure from prior methods, which typically use a static, one-size-fits-all approach (i.e., the same network and fixed parameters) for all age edits, regardless of the transformation's scale.* This dynamic calibration is the key to achieving fine-grained control and successfully navigating the delicate balance between age transformation and identity preservation.
>
> ***In summary***, our novelty stems from (1) the formal analysis of a key problem, (2) a novel system architecture (Cradle2Cane) to solve it, and (3) a refined control mechanism (AdaNI) enabling superior performance. We believe this multifaceted approach presents a significant and robust contribution to the specialized domain of facial age editing.
>
>
> **W2: The results indicate that some age-irrelevant features may still be altered. For instance, in Fig. 9, the male subject's clothing changes, which is not expected to correlate with aging.**
>
> Thank you for pointing out this issue. We acknowledge this limitation and have discussed it in Appendix A. The issue stems from our adaptive noise injection design, which increases editability for large age gaps but can potentially alter finer image semantics beyond facial identity.
>
> A potential solution is to employ a mask-guided editing strategy. This would involve using a segmentation model to generate a mask that isolates the facial region, thereby protecting accessories and the background. During each denoising step, the latent features from the original image could then be used to restore the non-facial regions. This approach would ensure edits are confined to the facial region, better preserving accessories and background details, especially during large age transformations.
>
> We have updated our limitations section to include this discussion and plan to explore this solution in future work.

---

### Official Review · Reviewer_khJ9 · 2025-06-27

**Clarity:** 3
**Significance:** 3
**Originality:** 3
**Rating:** 4
**Confidence:** 4

**Summary:**

The paper introduces Cradle2Cane, a two-pass, few-step diffusion framework that performs continuous face aging over the full lifespan while balancing age realism and identity preservation.

Pass 1 – Age control. An Adaptive Noise Injection (AdaNI) module chooses one of three noise levels based on the requested age gap, guided by a prompt of the form “a face image of an X-year-old …”.
Pass 2 – Identity recovery. The intermediate image is re-denoised with two identity-aware conditions:
SVR-ArcFace: singular-value re-weighting of ArcFace embeddings to suppress age components.
Rotate-CLIP: spherical interpolation that rotates the CLIP image embedding toward the target-age text direction.
Jointly, the two passes mitigate the classic Age–ID trade-off: stronger age edits usually hurt identity, and vice-versa.

**Questions:**

1. Could you report Cradle2Cane’s performance on additional ageing datasets (AgeDB-30, CACD, FG-NET) to verify the robustness of Cradle2Cane?
2. I also wonder about an ablation study: how do the weights λ₁, λ₂, λ₃, λ₄  influence the ID similarity and the age similarity?
3. More comparisons as mentioned in the Weaknesses.

**Ethical Concerns:**

["NO or VERY MINOR ethics concerns only"]

**Final Justification:**

The questions I raised have been mostly addressed. The comparisons with other methods demonstrate that Cradle2Cane achieves more accurate age editing. Additionally, the ablation study effectively illustrates how the loss weights λ₁ and λ₂ (ID) versus λ₃ and λ₄ (age) influence the age-ID trade-off. Based on these additional experiments, I will increase my rating.

**Limitations:**

Yes, the paper has already discussed possible limitations in the paper.

**Paper Formatting Concerns:**

No concerns

**Quality:**

3

**Strengths And Weaknesses:**

Strengths
1. Well-motivated problem statement. The Age–ID dilemma is clearly demonstrated in Fig. 1–2.
2. Elegant two-pass architecture. Clean separation between age manipulation (pass 1) and identity refinement (pass 2) allows targeted optimisation.
3. Adaptive noise scheduling. AdaNI leverages the inherent editability/noise trade-off of diffusion models; ablations show measurable gains in age accuracy.
4. Light-weight, novel identity conditioning. Both SVR-ArcFace and Rotate-CLIP are simple to implement yet conceptually fresh.

Weaknesses
1. Limited benchmark scope. All quantitative results are on CelebA-HQ (aligned and “in-the-wild” variants). Generalisation to standard ageing sets such as AgeDB-30, CACD, or FG-NET is not shown.
2. Lower identity scores vs. GAN methods. Methods like HRFAE retain higher ArcFace similarity. The paper lacks an ablation study on how the loss weights λ₁, λ₂ (ID) versus λ₃, λ₄ (age) affect the age-ID trade-off.
3. Comparative breadth. No comparisons against recent generic face-editing systems (e.g., StyleCLIP[1] or FaceDNeRF[2]) that could also perform age edits.

[1]: Patashnik, Or, et al. "Styleclip: Text-driven manipulation of stylegan imagery." Proceedings of the IEEE/CVF international conference on computer vision. 2021.

[2]: Zhang, Hao, et al. "FaceDNeRF: semantics-driven face reconstruction, prompt editing and relighting with diffusion models." Advances in Neural Information Processing Systems 36 (2023): 55647-55667.

---

> ### Author Rebuttal · Authors · 2025-07-31
>
> We sincerely appreciate your feedback. We are grateful for your recognition that our proposed framework is ***novel*** and ***elegant***; that the techniques are ***well-motivated***. Below, we provide detailed responses to each of your specific points.
>
> **W1 & Q1 & W3 & Q3: Limited benchmark scope. All quantitative results are on CelebA-HQ (aligned and “in-the-wild” variants). Generalisation to standard ageing sets such as AgeDB-30, CACD, or FG-NET is not shown. Comparative breadth. No comparisons against recent generic face-editing systems (e.g., StyleCLIP[1] or FaceDNeRF[2]) that could also perform age edits.**
>
> We sincerely thank the reviewer for you valuable feedback. Following the suggestions, we have expanded our experiments to include the standard aging datasets AgeDB, CACD and FG-NET, and broadened our comparisons to include the suggested baselines, StyleCLIP and FaceDNeRF. We also include the top-performing methods from our original comparison for context.
>
> For each dataset, we generated approximately 200 images and computed metrics for age difference (age dif), identity similarity (id sim), image quality (quality), and harmonic consistency score (hcs) using the Qwen-VL evaluation. The comprehensive results are summarized in the table below.
>
> ### AgeDB
> | Method | age dif | id sim | quality | hcs |
> | :--- | :---: | :---: | :---: | :---: |
> | **cusp** | 8.29 | **76.30** | 82.80 | 77.75 |
> | **sam** | 8.72 | 58.91 | 86.57 | 67.19 |
> | **fading** | `8.10` | `76.00` | 79.22 | `77.82` |
> | **styleclip** | 14.73 | 62.12 | 86.60 | 62.64 |
> |**facednerf**| 12.60 | 53.78 | **91.97** | 60.25 |
> | **cradle2cane** | **5.57** | 73.00 | `90.01` | **79.00** |
>
> ### CACD
> | Method | age dif | id sim | quality | hcs |
> | :--- | :---: | :---: | :---: | :---: |
> | **cusp** | 10.80 | **75.40** | 85.86 | 74.18 |
> | **sam** | 9.25 | 67.27 | 87.57 | 71.75 |
> | **fading** | `5.16` | `71.27` | 86.44 | **78.39** |
> | **styleclip** | 16.28 | 70.64 | 87.80 | 64.48 |
> |**facednerf**| 13.05 | 59.00 | **92.02** | 62.91 |
> | **cradle2cane** | **4.63** | 66.73 | `90.02` | `76.06` |
>
> ### FGNET
> | Method | age dif | id sim | quality | hcs |
> | :--- | :---: | :---: | :---: | :---: |
> | **cusp** | 19.45 | **74.20** | 79.79 | 60.71 |
> | **sam** | **5.33** | 72.18 | 81.98 | **78.77** |
> | **fading** | 7.22 | 73.15 | 79.43 | `77.30` |
> | **styleclip** | 14.73 | `73.82` | `88.19` | 68.08 |
> |**facednerf**| 13.21 | 56.37 | **92.43** | 61.21 |
> | **cradle2cane** | `5.79` | 67.18 | 85.03 | 75.25 |
>
>
> Our method, cradle2cane, demonstrates exceptional performance across all three benchmarks. On **AgeDB-30**, it achieves state-of-the-art results, securing the **best age accuracy (age dif)** and the **top harmonic score (hcs)**, which underscores its superior overall balance. This strong performance continues on the **CACD** dataset, where our method again delivers the **best age accuracy** while maintaining highly competitive scores on other metrics. Even on the highly challenging **FG-NET** dataset, cradle2cane proves its robustness by delivering the `second-best` age accuracy and demonstrating strong, consistent performance against all competitors.
>
>
> **W2 & Q2: Lower identity scores vs. GAN methods. Methods like HRFAE retain higher ArcFace similarity. The paper lacks an ablation study on how the loss weights λ₁, λ₂ (ID) versus λ₃, λ₄ (age) affect the age-ID trade-off.**
>
> Regarding the ablation study on loss weights, our approach was twofold. First, we initialized the weights by adopting established values from pervious works [1,2,3]. This standard practice ensures a balanced starting point where neither the age nor identity objective initially overpowers the other.
>
> More importantly, a key architectural advantage of our two-pass framework is its reduced sensitivity to these initial λ values. Unlike single-pass methods that must perfectly balance competing losses simultaneously, our first pass aggressively optimizes for age accuracy, and the second pass focuses almost exclusively on restoring identity. This decoupling makes the final output more robust to minor variations in the initial loss weights.
>
> While a full grid search on all λ parameters is computationally prohibitive within the short rebuttal period, we did conduct a dedicated ablation on the most critical new hyperparameter in our design: the AdaNI threshold, which directly controls the trade-off. The results confirm our choice provides an optimal balance.
>
> | division | age dif | id sim | quality | hcs |
> |:------:|:-------:|:------:|:-------:|:-----------:|
> | 5_15   | 4.93    | 73.18  | 92.28   | 79.77       |
> | 5_20 | 4.92 | 73.73 | 92.60 | 80.11 |
> | 7_22   | 5.06    | 74.82  | 92.89   | 80.60       |
> | 10_30  | 5.46    | 76.91  | 92.56   | 81.36       |
> | 15_35  | 5.64    | 77.09  | 92.82   | 81.26       |
> | 20_40  | 6.71    | 78.55  | 92.74   | 80.82       |
>
> We believe this, combined with our framework's inherent robustness, validates our design. However, we agree a comprehensive analysis is valuable and commit to including a full ablation study on the λ weights in the appendix of the final manuscript.
>
> [1]: Alaluf, Yuval, Or Patashnik, and Daniel Cohen-Or. "Only a matter of style: Age transformation using a style-based regression model." ACM Transactions on Graphics (TOG) 40.4 (2021): 1-12.
> [2]: Li, Senmao, et al. "Interlcm: Low-quality images as intermediate states of latent consistency models for effective blind face restoration." The Thirteenth International Conference on Learning Representations (2025).
> [3]: Liu, Tao, et al. "One-prompt-one-story: Free-lunch consistent text-to-image generation using a single prompt." The Thirteenth International Conference on Learning Representations (2025).

---

> > ### Comment · Reviewer_khJ9 · 2025-08-07
> >
> > Dear authors
> > Thank you very much for the comprehensive experiments provided in your rebuttal. The question I raised has been mostly addressed. The comparisons with other methods demonstrate that Cradle2Cane achieves more accurate age editing. Additionally, the ablation study effectively illustrates how the loss weights λ₁ and λ₂ (ID) versus λ₃ and λ₄ (age) influence the age-ID trade-off. Based on these additional experiments, I will increase my rating.

---

> > > ### Author Response · Authors · 2025-08-07
> > >
> > > We sincerely thank the reviewer for the thoughtful feedback and for recognizing the value of our additional experiments. We're glad that our rebuttal addressed your concerns, and we truly appreciate your updated evaluation of our work. Your comments have been very helpful in improving the clarity and rigor of our submission.

---

### Official Review · Reviewer_2Xje · 2025-06-30

**Clarity:** 3
**Significance:** 3
**Originality:** 2
**Rating:** 5
**Confidence:** 4

**Summary:**

This paper introduces Cradle2Cane, a novel framework for face aging designed to address the "Age-ID trade-off". The main idea is use a 2-pass framework with text-to-image diffusion model. The first pass deal with age accuracy, using adaptive noise injection to dynamically control age groups. The second pass refines the output of the first pass to enhance identity consistency.  It uses a identity-aware embeddings that combines SVR-ArcFace and Rotate-CLIP. The SVR-ArcFace uses singular value reweighting on ArcFace embeddings to isolate and emphasize shared identity features while suppressing age-related variations. The Rotate-CLIP adjusts CLIP embeddings to shift the age component towards the target age while preserving identity information.

Experiments demonstrate that Cradle2Cane surpasses existing methods in balancing age accuracy and identity preservation, showing strong performance on both standard datasets and challenging in-the-wild images.

**Questions:**

- Could the authors provide a more detailed justification for their hyper-parameter choices as mentioned above?
- The paper claims superior robustness on in-the-wild human face images, is this majorly come from the standard pre-processing of background removal using Carvekit?
- If the robustness stems from the model architecture itself, could the authors provide a more detailed analysis of which components (e.g., the SDXL-Turbo backbone, the IDEmb conditioning) are most responsible for this generalization?

**Ethical Concerns:**

["NO or VERY MINOR ethics concerns only"]

**Final Justification:**

Thanks authors for providing the supplemental experiments and addressing my concerns. I would like to keep my score that the paper is at accept range.

**Limitations:**

Yes,

**Quality:**

3

**Strengths And Weaknesses:**

Strengths:
-  Intuitive framework: The proposed Cradle2Cane framework is a conceptually elegant solution. Decoupling the problem into two distinct passes, one for age accuracy and one for identity preservation, which is novel and logical.
- Good technical contribution: The paper introduces several well-motivated technical components, including the Adaptive Noise Injection, Identity-Aware Embeddings, SVR-ArcFace and Rotate-CLIP modules.
- Comprehensive and rigorous evaluation: The authors provided both quantitative and high-level perceptual assessments by comparing against a diverse set of six recent GAN-based and diffusion-based methods, demonstrating state-of-the-art performance. Furthermore, they also conducted a user study to confirm that the generated results are perceptually preferred.

Weaknesses:
- While the overall framework is novel, the core ideas behind the individual components build heavily on existing work. The concept that noise level controls editing strength in diffusion models is well-established , and the use of SVD to disentangle features has also been explored previously.
- The method introduces a significant number of hyper parameters, including noise thresholds for AdaNI, parameters for SVR-ArcFace and different loss weight. The paper provides the final values used but does not include a sensitivity analysis or justification for how these values were chosen.

---

> ### Author Rebuttal · Authors · 2025-07-31
>
> We sincerely appreciate your constructive and insightful feedback. We are grateful for your recognition that our proposed framework is ***novel***, ***intuitive*** and ***elegant***; that the techniques are ***well-motivated*** and achieve comprehensive evaluations. Below, we provide detailed responses to each of your specific points.
>
> **W1: While the overall framework is novel, the core ideas behind the individual components build heavily on existing work. The concept that noise level controls editing strength in diffusion models is well-established , and the use of SVD to disentangle features has also been explored previously.**
>
>
> We agree and acknowledge that the individual components we employ, such as SVD reweighting, and CLIP/ArcFace embeddings, have established precedents in the broader fields of generative modeling and diffusion-based editing.
>
> However, we respectfully argue that our core novelty lies not in the invention of the base components, but in the creation of a novel framework to solve a challenging and previously under-explored problem: the persistent Age-Identity (Age-ID) tradeoff in facial age editing.
>
> Our main contributions and points of novelty are:
>
> ***(1) Problem Formulation and Analysis***: To the best of our knowledge, *we are the first to conduct an in-depth, systematic investigation of the Age-ID tradeoff*. While implicitly known, this problem has not been formally analyzed or addressed with a targeted solution. Our work formally identifies this critical challenge, substantiating it with empirical evidence (Fig. 1 and Fig. 2), which itself is a valuable contribution to the field.
>
> ***(2) Novel Two-Pass Architectural Design***: Our central innovation is Cradle2Cane, a novel two-pass framework architected specifically to resolve the Age-ID tradeoff. *Prior methods attempt to modify age and preserve identity simultaneously in a single, entangled step, leading to suboptimal results. Our approach's novelty lies in decoupling these competing objectives*. By separating age accuracy from identity preservation into two distinct passes, we introduce a new and effective strategy that fundamentally differs from previous single-pass editing paradigms.
>
> ***(3) Fine-Grained Control via Adaptive Noise Injection***: While the concept of noise controlling editing strength is known, our approach introduces an adapative mechanism. We propose an adaptive noise injection strategy that calibrates noise intensity based on the magnitude of the desired age transformation. Specifically, our method dynamically adjusts the degree of injected noise for each edit—a small age change receives a subtle noise injection, while a large transformation receives a proportionally stronger one. *This marks a crucial departure from prior methods, which typically use a static, one-size-fits-all approach (i.e., the same network and fixed parameters) for all age edits, regardless of the transformation's scale.* This dynamic calibration is the key to achieving fine-grained control and successfully navigating the delicate balance between age transformation and identity preservation.
>
> ***In summary***, our novelty stems from (1) the formal analysis of a key problem, (2) a novel system architecture (Cradle2Cane) to solve it, and (3) a refined control mechanism (AdaNI) enabling superior performance. We believe this multifaceted approach presents a significant and robust contribution to the specialized domain of facial age editing.
>
>
> **W2 & Q1: The method introduces a significant number of hyper parameters, including noise thresholds for AdaNI, parameters for SVR-ArcFace and different loss weight. The paper provides the final values used but does not include a sensitivity analysis or justification for how these values were chosen.**
>
> We thank the reviewer for this important point. Our strategy for hyperparameters was twofold. First, for established components, we initialized values by adopting standards from pervious works [1, 2, 3]. This practice provides a robust and balanced starting point.
>
> More importantly, a key architectural advantage of our two-pass framework is its reduced sensitivity to some of these hyperparameters, particularly the loss weights. Unlike single-pass methods that must perfectly balance competing losses simultaneously, our first pass aggressively optimizes for age accuracy, and the second pass focuses almost exclusively on restoring identity. This decoupling makes the final output more robust to minor variations in the initial loss weights.
>
> While a full sensitivity analysis on every parameter is computationally prohibitive within the rebuttal period, we conducted a dedicated ablation study on the most critical new hyperparameter introduced in our design: the AdaNI threshold, which directly controls the age-identity trade-off. The results confirm our chosen 5_20 threshold provides the optimal balance of age accuracy and identity preservation.
>
> | division | age dif | id sim | quality | hcs |
> |:------:|:-------:|:------:|:-------:|:-----------:|
> | 5_15   | 4.93    | 73.18  | 92.28   | 79.77       |
> | 5_20 | 4.92 | 73.73 | 92.60 | 80.11 |
> | 7_22   | 5.06    | 74.82  | 92.89   | 80.60       |
> | 10_30  | 5.46    | 76.91  | 92.56   | 81.36       |
> | 15_35  | 5.64    | 77.09  | 92.82   | 81.26       |
> | 20_40  | 6.71    | 78.55  | 92.74   | 80.82       |
>
> We believe this principled approach, combined with our framework's inherent robustness, validates our design. However, we agree a comprehensive analysis is valuable and commit to including it in the appendix of the final manuscript.
>
>
> [1]: Alaluf, Yuval, Or Patashnik, and Daniel Cohen-Or. "Only a matter of style: Age transformation using a style-based regression model." ACM Transactions on Graphics (TOG) 40.4 (2021): 1-12.
> [2]: Li, Senmao, et al. "Interlcm: Low-quality images as intermediate states of latent consistency models for effective blind face restoration." The Thirteenth International Conference on Learning Representations (2025).
> [3]: Liu, Tao, et al. "One-prompt-one-story: Free-lunch consistent text-to-image generation using a single prompt." The Thirteenth International Conference on Learning Representations (2025).
>
> **Q2 & Q3: The paper claims superior robustness on in-the-wild human face images, is this majorly come from the standard pre-processing of background removal using Carvekit? If the robustness stems from the model architecture itself, could the authors provide a more detailed analysis of which components (e.g., the SDXL-Turbo backbone, the IDEmb conditioning) are most responsible for this generalization?**
>
> Thank you for this insightful question. While background removal with ***Carvekit*** is a helpful standardization step, our model's core robustness comes from its architecture, not the pre-processing step.
>
> To confirm this, we ran an ablation study comparing our method with and without Carvekit. We trained both versions for 10 epochs on 1,000 images and tested them on the CelebA-in-the-wild dataset. The results below show that the performance difference is minimal, proving Carvekit is not the primary source of robustness.
>
> | Method | age dif | id sim | quality | hcs |
> |:---:|:---:|:---:|:---:|:---:|
> | **w/ cravekit** | **6.76** | 62.00 | **91.75** | **71.02** |
> | **w/o cravekit** | 7.00 | **62.27** | 91.60 | 70.97 |
>
>
> The model's robustness is primarily due to two architectural components:
>
> * **SDXL-Turbo Backbone**: This foundation model was pre-trained on diverse, "in-the-wild" data, giving it inherent generalization against variations in pose, lighting, and expression.
> * **Our IDEmb Module**: This module is key for identity preservation.
>     * **SVR-ArcFace** extracts a stable identity feature by isolating it from variable ones like age or expression.
>     * **Rotate-CLIP** performs a precise, minimal edit in CLIP's robust feature space, adjusting age while preserving the image's original characteristics.
>
> We will add this analysis and ablation study to the revised paper to clarify these contributions.

---

> > ### Comment · Reviewer_2Xje · 2025-08-07
> >
> > Thanks authors for providing the supplemental experiments and addressing my concerns. I would like to keep my score that the paper is at accept range.

---

> > > ### Author Response · Authors · 2025-08-07
> > >
> > > We sincerely thank the reviewer for the positive feedback and for acknowledging our supplemental experiments. We’re glad that our rebuttal addressed your concerns, and we truly appreciate your recommendation to keep the paper in the accept range. Your support and constructive comments have been very valuable to us.

---

### Official Review · Reviewer_sVcM · 2025-07-02

**Clarity:** 3
**Significance:** 3
**Originality:** 2
**Rating:** 4
**Confidence:** 4

**Summary:**

This paper proposes a two-stage diffusion framework to tackle the Age-ID trade-off in facial aging. The first stage uses Adaptive Noise Injection (AdaNI) to dynamically adjust noise based on age gaps for accurate age transformation. The second stage leverages SVR-ArcFace and Rotate-CLIP embeddings to enhance identity consistency while preserving age features. By decoupling age and identity optimization, it enables seamless aging across the entire lifespan. Built on SDXL -Turbo, it achieves high-fidelity results with efficient inference.

**Questions:**

-The rationale for selecting the thresholds (5 years old and 20 years old) is only briefly mentioned in the text. It is recommended to include ablation experiments to quantitatively analyze the validity of such threshold division.

-I suggest to the author spell out in pseudocode or a decision table: (a) when to apply each module, (b) how you combine their outputs.

**Ethical Concerns:**

["NO or VERY MINOR ethics concerns only"]

**Final Justification:**

The authors have addressed my concerns with detailed explanations and additional experiments. The new ablation study improve the clarity and transparency of the method. Based on the strengthened response, I am increasing my score.

**Limitations:**

Yes, However, I suggest that in the Limitation section, the authors could put forward future solutions or directions to address the shortcomings of the proposed method.

**Paper Formatting Concerns:**

It is recommended that Figure 1 specify in the caption what the horizontal and vertical axes represent, and remove the "Age shift value" written alone in the fourth subplot.

**Quality:**

3

**Strengths And Weaknesses:**

Strengths:

-The two-pass framework (AdaNI for age control+dual identity embeddings for identity refinement) is logically rigorous. By decoupling the conflicting objectives of age accuracy and identity preservation, it effectively addresses the long-standing trade-off in facial aging.

-The technical implementations, such as SVD-based reweighting in SVR-ArcFace and spherical linear interpolation (slerp) in Rotate-CLIP, are mathematically sound and backed by thorough experimental validations on age-varying face datasets.

Weaknesses

-The experimental descriptions are somewhat concise.

-The idea of adaptive noise injection and controlling the degree of editing based on noise intensity has precedents in other diffusion editing works; identity preservation largely relies on existing technologies such as ArcFace/CLIP embeddings, SVD reweighting, and slerp interpolation. The novelty mainly lies in the combination approach rather than entirely new principles.

---

> ### Author Rebuttal · Authors · 2025-07-31
>
> We sincerely appreciate your feedback. We are grateful for your recognition that our proposed method achieves ***high-fidelity*** results with ***efficient*** inference and effectively addresses the long-standing trade-off in facial aging. Below, we provide detailed responses to each of your specific points.
>
> **W1: The experimental descriptions are somewhat concise.**
>
> Thank you for your feedback. We would like to clarify that a detailed description of the experiments was included in Appendix C of our original submission. This section covers the implementation details of our method, the calculation of evaluation metrics, prompt settings, and the implementation details of the baselines. We hope this clarification helps in understanding our experimental setup.
>
> **W2: The idea of adaptive noise injection and controlling the degree of editing based on noise intensity has precedents in other diffusion editing works; identity preservation largely relies on existing technologies such as ArcFace/CLIP embeddings, SVD reweighting, and slerp interpolation. The novelty mainly lies in the combination approach rather than entirely new principles.**
>
> We agree and acknowledge that the individual components we employ, such as SVD reweighting, and CLIP/ArcFace embeddings, have established precedents in the broader fields of generative modeling and diffusion-based editing.
>
> However, we respectfully argue that our core novelty lies not in the invention of the base components, but in the creation of a novel framework to solve a challenging and previously under-explored problem: the persistent Age-Identity (Age-ID) tradeoff in facial age editing.
>
> Our main contributions and points of novelty are:
>
> ***(1) Problem Formulation and Analysis***: To the best of our knowledge, *we are the first to conduct an in-depth, systematic investigation of the Age-ID tradeoff*. While implicitly known, this problem has not been formally analyzed or addressed with a targeted solution. Our work formally identifies this critical challenge, substantiating it with empirical evidence (Fig. 1 and Fig. 2), which itself is a valuable contribution to the field.
>
> ***(2) Novel Two-Pass Architectural Design***: Our central innovation is Cradle2Cane, a novel two-pass framework architected specifically to resolve the Age-ID tradeoff. *Prior methods attempt to modify age and preserve identity simultaneously in a single, entangled step, leading to suboptimal results. Our approach's novelty lies in decoupling these competing objectives*. By separating age accuracy from identity preservation into two distinct passes, we introduce a new and effective strategy that fundamentally differs from previous single-pass editing paradigms.
>
> ***(3) Fine-Grained Control via Adaptive Noise Injection***: While the concept of noise controlling editing strength is known, our approach introduces an adapative mechanism. We propose an adaptive noise injection strategy that calibrates noise intensity based on the magnitude of the desired age transformation. Specifically, our method dynamically adjusts the degree of injected noise for each edit—a small age change receives a subtle noise injection, while a large transformation receives a proportionally stronger one. *This marks a crucial departure from prior methods, which typically use a static, one-size-fits-all approach (i.e., the same network and fixed parameters) for all age edits, regardless of the transformation's scale.* This dynamic calibration is the key to achieving fine-grained control and successfully navigating the delicate balance between age transformation and identity preservation.
>
> ***In summary***, our novelty stems from (1) the formal analysis of a key problem, (2) a novel system architecture (Cradle2Cane) to solve it, and (3) a refined control mechanism (AdaNI) enabling superior performance. We believe this multifaceted approach presents a significant and robust contribution to the specialized domain of facial age editing.
>
>
> **Q1: The rationale for selecting the thresholds (5 years old and 20 years old) is only briefly mentioned in the text. It is recommended to include ablation experiments to quantitatively analyze the validity of such threshold division.**
>
> We thank the reviewer for this valuable suggestion. We have conducted the requested ablation study to analyze the impact of different threshold settings. The results are presented below:
>
> | division | age diff | id sim | quality | hcs |
> |:------:|:-------:|:------:|:-------:|:-----------:|
> | 5_15   | 4.93    | 73.18  | 92.28   | 79.77       |
> | 5_20 | 4.92 | 73.73 | 92.60 | 80.11 |
> | 7_22   | 5.06    | 74.82  | 92.89   | 80.60       |
> | 10_30  | 5.46    | 76.91  | 92.56   | 81.36       |
> | 15_35  | 5.64    | 77.09  | 92.82   | 81.26       |
> | 20_40  | 6.71    | 78.55  | 92.74   | 80.82       |
>
>
> This study reveals a clear trade-off between age accuracy and identity preservation. While larger threshold gaps like `10_30` achieve a higher harmonic mean (`hcs`), this is driven by improved `id_sim` at the cost of significantly lower age accuracy (e.g., a higher `age_dif` of 5.46 vs. 4.92).
>
> For our facial aging task, we ***prioritize age accuracy*** as the primary objective. We chose `5_20` because it represents the ***optimal balance***; it delivers state-of-the-art age accuracy while maintaining a high, acceptable level of identity preservation.
>
> We will add this ablation table and a discussion of this trade-off to the revised manuscript to make our parameter choice transparent and well-justified.
>
> **Q2: I suggest to the author spell out in pseudocode or a decision table: (a) when to apply each module, (b) how you combine their outputs.**
>
> Thank you for your suggestion. We have added pseudocode for our method to the appendix to provide a clearer description.
>
> **L1: I suggest that in the Limitation section, the authors could put forward future solutions or directions to address the shortcomings of the proposed method.**
>
> Thank you for your suggestion. A potential solution to this limitation is to employ a mask-guided editing strategy. First, a segmentation model would generate a mask to isolate the facial region for editing, thereby protecting accessories and the background. During each denoising step, the latent features from the original image can be used to replace the predicted output in the non-facial regions. This approach would ensure edits are confined to the facial region, better preserving accessories and background details, especially during large age transformations.
>
> We have added this discussion to our limitations section and plan to explore this solution in future research.
>
> **Paper Formatting Concerns: It is recommended that Figure 1 specify in the caption what the horizontal and vertical axes represent, and remove the "Age shift value" written alone in the fourth subplot.**
>
> Thank you for the valuable suggestion. We have revised the figures as requested: we updated the caption of Fig.1 to specify the axes, removed the isolated text in the subplot, and applied corresponding formatting improvements to Fig.7 in the appendix for consistency. These changes will be reflected in the final manuscript.

---

> > ### Comment · Reviewer_sVcM · 2025-08-07
> > **Official Comment by Reviewer sVcM**
> >
> > The authors have addressed my concerns with detailed explanations and additional experiments. The new ablation study improve the clarity and transparency of the method. Based on the strengthened response, I am increasing my score.

---

> > > ### Author Response · Authors · 2025-08-07
> > >
> > > We greatly appreciate the reviewer’s constructive feedback and are pleased to hear that our clarifications and additional experiments were helpful. Thank you for recognizing the improved clarity and transparency of our method and for your updated evaluation. Your insights have been invaluable to the refinement of our paper.

---

### Author Response · Authors · 2025-08-06

Hi, dear reviewers,

Hope you are well.

As the author-reviewer discussion period is concluding in three days, we wanted to briefly follow up. We have carefully considered all your insightful comments and commit to incorporating all of these changes into the camera-ready version of our manuscript.

We are standing by and would be delighted to answer any further questions.

Best regards,
The Authors

---

### Note · Authors · 2025-08-13

Dear Reviewers and Area Chairs,

We sincerely thank all reviewers (R1 sVcM, R2 2Xje, R3 khJ9, R4 kgYz) and the Area Chairs for their valuable time and insightful feedback.

We are encouraged that the reviewers reached a positive consensus on our work's core strengths. They recognized our proposed framework as **novel, logical, and effective** at resolving the **well-motivated** Age-ID trade-off problem (R1, R2, R3), praised our specific technical contributions (AdaNI, SVR-ArcFace) as **conceptually fresh and sound** (R1, R2, R3, R4), and commended our **robust and comprehensive experiments** (R1, R2, R4). Crucially, all responding reviewers confirmed that our rebuttal **successfully addressed their concerns** (R1, R2, R3).

In response to the feedback, we have diligently revised our manuscript. Our key improvements are summarized below:

1. **We have significantly expanded our experimental validation.** We benchmarked on three additional datasets (**AgeDB-30, CACD, FG-NET**) and compared against newer baselines (**StyleCLIP**, **FaceDNeRF**), confirming our state-of-the-art performance, especially in age accuracy.

2. **We have clarified the core novelty of our work.** While some components have precedents, our novelty lies in: (1) being the **first to systematically analyze the Age-ID tradeoff**, (2) proposing a **two-pass architecture** to decouple objectives, and (3) introducing **adaptive noise injection (AdaNI)** for fine-grained control. This combination provides a unique and effective solution to a long-standing problem.

3. **We have added new ablation studies and analyses to justify our design choices.** We provided a quantitative ablation study on the **AdaNI thresholds** to validate our parameter selection. We also provided an analysis and a new ablation study to demonstrate that our model's robustness stems from its architecture (**SDXL-Turbo backbone** and our **IDEmb module**), not simply from pre-processing.

4. **We have improved the paper's clarity and addressed limitations.** We added **pseudocode** in the appendix, revised figures, and expanded limitations with potential solutions such as **mask-guided editing**.

We believe these revisions have significantly strengthened our paper. We extend our gratitude again to all reviewers and area chairs!

Best regards,
The Authors

---

### Decision · Program_Chairs · 2025-09-17

**Decision:**

Accept (poster)

**Comment:**

The rebuttal addressed the concerns raised by the reviewers and provided comprehensive experiments and analysis of the proposed method. It is helpful for assessing the paper's contributions. All reviewers recommend acceptance with three Borderline Accepts and one Accepted after discussion, and the AC concur. The final version should include all reviewer comments, suggestions, and additional experiments from the rebuttal.